



# Seismicity during and after stimulation of a 6.1 km deep Enhanced Geothermal System in Helsinki, Finland

Maria Leonhardt[1], Grzegorz Kwiatek[1], Patricia Martínez-Garzón[1], Marco Bohnhoff[1,2], Tero Saarno[3], Pekka Heikkinen[4], and Georg Dresen[1,5]

[1]Helmholtz Centre Potsdam, GFZ German Research Centre for Geosciences, Section 4.2: Geomechanics and Scientific Drilling, Potsdam, Germany, [2]Department of Earth Sciences, Free University of Berlin, Berlin, Germany, [3]St1 Deep Heat Oy, Helsinki, Finland, [4]Department of Geosciences and Geography, University of Helsinki, Helsinki, Finland, [5]Institute of Earth and Environmental Sciences, University of Potsdam, Potsdam, Germany

*Correspondence to*: Maria Leonhardt (maria.leonhardt@gfz-potsdam.de)

**Abstract.** In this study, we present a high-resolution dataset of seismicity framing the stimulation campaign of a 6.1 km deep Enhanced Geothermal System (EGS) in Helsinki suburban area and discuss the complexity of fracture network development. Within St1 Deep Heat project, 18,160 m$^3$ of water was injected over 49 days in summer 2018. The seismicity was monitored by a seismic network of near-surface borehole sensors framing the EGS site

in combination with a multi-level geophone array located at ≥2 km depth. We expand the original catalog of Kwiatek et al. (2019) and provide the community with the dataset including detected seismic events and earthquakes that occurred two month after the end of injection, totalling to 61,163 events. We relocated events of the catalog with sufficient number of available phase onsets and moment magnitudes between $M_W$ -0.7 and $M_W$ 1.9 using the double-difference technique and a new velocity model derived from a post-stimulation vertical seismic

profiling campaign. The analysis of the fault network development at reservoir depth of 4.5-7 km is one primary focus of this study. To achieve this, we investigate 191 focal mechanisms of the induced seismicity using cross-correlation based technique. Our results indicate that seismicity occurred in three spatially separated clusters centered around the injection well. We observe a spatio-temporal migration of the seismicity during the stimulation starting from the injection well in northwest (NW) - southeast (SE) direction and in northeast (NE) direction

towards greater depth. The spatial evolution of the cumulative seismic moment, the distribution of events with $M_W \geq 1$ and the fault plane orientations of focal mechanisms indicate an active network of at least three NW-SE to NNW-SSE orientated permeable zones which is interpreted to be responsible for migration of seismic activity away from the injection well. Fault plane solutions of the best-constrained focal mechanisms as well as results for the local stress field orientation indicate a reverse faulting regime and suggest that seismic slip occurred on a sub-





parallel network of pre-existing weak fractures favorably oriented with the stress field, striking NNW-SEE with a
dip of 45° ENE, parallel to the injection well.

## 1    Introduction

Deep geothermal energy is considered as a potential source of low $CO_2$-emission energy to replace fossil fuels.
The successful development of deep geothermal reservoirs is crucial for the economic production of hot fluids for
energy production. However, crystalline basement rocks hosting deep geothermal reservoirs in general are low-
porosity and low-permeability formations. In Enhanced Geothermal Systems (EGS) hydraulic stimulation with
massive fluid injection is applied to improve reservoir permeability (e.g. Giardini, 2009). Fluid injection at depth
in EGS stimulations and in waste-water disposal is commonly associated with induced seismicity (e.g. Ellsworth,
2013; Majer et al., 2012). Successful mitigation of induced seismic hazard is important for public acceptance of
geothermal projects as significant concern exists related to the occurrence of larger induced earthquakes during
previous EGS projects, e.g. in Basel and St. Gallen, Switzerland (e.g. Giardini, 2009; Diehl et al., 2017) or most
recently in Pohang, South Korea (Hofmann et al., 2019; Ellsworth et al., 2019).

A well-designed seismic network is pre-requisite for high-resolution data acquisition, real-time seismic
monitoring and analysis of induced seismicity (e.g. Bohnhoff et al., 2018). Subsequent feeding of seismic data into
a traffic-light-system (TLS) may substantially contribute to mitigate the associated seismic hazard and risk. A
successful and safe approach to stimulation of the world's deepest EGS in the metropolitan area of Helsinki was
recently presented by Kwiatek et al. (2019). Over 49 days in summer 2018, the St1 Deep Heat Company injected
more than 18,000 m³ of water at 6.1 km depth. A $M_W$ 2.1 red alert threshold of the TLS defined by the local
authorities was successfully avoided by a careful adjustment of the hydraulic energy input in response to real-time
monitoring of the spatio-temporal evolution of seismicity. The largest seismic event was confined to a moment
magnitude of $M_W$ 1.9 (Ader et al., 2019; Kwiatek et al., 2019).

High quality state-of-the art analysis of induced seismic waveform data is crucial for a detailed reservoir
characterization (Kwiatek et al., 2013). High precision locations of hypocenters are typically obtained by applying
relocation techniques such as the double-difference method (Waldhauser and Ellsworth, 2000). Using relocated
data, a precise spatio-temporal evolution of induced seismicity can be tracked providing insight in fluid migration
pathways in the reservoir (e.g. Kwiatek et al., 2015; Diehl et al., 2017). In addition, seismic source parameters
such as seismic moment and source size provide crucial insights into the fracture network geometry.

Bentz et al. (2020) recently showed that many EGS fluid injections display an extended period of stable
evolution of the cumulative seismic moment. Following Galis et al. (2015), this indicates the growth of self-





arrested ruptures, in contrast to unstable increase of seismic moment resulting in runaway ruptures that are only limited by the size of tectonic faults. Thus, unusual trends or potential changes in the seismic moment evolution may provide information on growth and activation of ruptures and thus also on the anthropogenic seismic hazard and subsequent risk. For example, Bentz et al. (2020) observed a steep and not stabilizing increase of the cumulative seismic moment potentially signifying unbound rupture propagation during stimulation for the Pohang

EGS project. Dynamic source characteristics of seismic events including radiated energy, stress drop and apparent stress allow evaluating seismic injection efficiency (Maxwell, 2008) and estimating energy budget of a stimulation campaign. Moreover, focal mechanisms provide important information for hazard assessment, as they can illuminate activation of large pre-existing structures such as major and potentially critically pre-stressed faults (e.g. Deichmann and Giardini, 2009; Ellsworth et al., 2019). Using focal mechanisms, Ellsworth et al. (2019) showed

that induced seismicity activated a fault zone which ultimately triggered the large $M_W$ 5.5 earthquake at Pohang. The authors suggested that seismic analysis performed during stimulation sequences may provide early information on increasing seismic hazard. In addition, stress tensor inversion of focal mechanism data using e.g. the *MSATSI* (Martínez-Garzón et al., 2014) or *BRMT* (D'Auria and Massa, 2015) approaches allow estimating potential changes of the local stress field but require high-quality seismic waveform data from dense local seismic

networks. Studying spatial and temporal variations of the stress field orientation contribute to understanding complex seismo-mechanical processes occurring in the reservoir during injection (Kwiatek et al., 2013). Martínez-Garzón et al. (2013) first observed a clear correlation of temporal stress changes in response to high injection rates at The Geysers geothermal field.

In this study we present a refined high-resolution dataset of seismicity induced during stimulation of the

world's deepest geothermal EGS in the Helsinki suburban area in 2018 (Kwiatek et al., 2019; Ader et al., 2019; Hillers et al., 2020). The data was collected using a combined seismic network of individual sensors in shallow boreholes framing the injection site combined with a multi-level vertical geophone array at ≥2 km depth. Our dataset expands, refines and completes the original study of Kwiatek et al. (2019). We include seismic events which occurred after the end of the hydraulic stimulation and refine the seismic catalog using double-difference

relocation with a new derived velocity model from a post-injection Vertical Seismic Profiling (VSP) campaign. To analyze the structural complexity of the reservoir, we investigate the spatio-temporal seismicity evolution and the temporal as well as spatial distribution of the seismic moment release during and after stimulation. This analysis is supported by an extensive catalog of source mechanisms derived from a cross-correlation based technique. Information on the local stress field orientation is derived from seismicity data. We discuss the evolution of

potentially permeable zones in the reservoir and the re-activation of a network of small-scale fractures during and after stimulation.





## 2    Methodology

### 2.1 Seismic catalog reprocessing

Expanding the study of Kwiatek et al. (2019), we enhanced, reprocessed and relocated the original seismic catalog
now also including post-injection events between July 22$^{nd}$ and September 24$^{th}$. During and after the stimulation, induced seismicity was monitored by a dense seismic network of three-component sensors consisting of a 12-level vertical borehole array as well as 12 near-surface seismometers with full azimuthal coverage. The borehole array with 15 Hz sensors, sampled at 2 kHz, was installed at a depth from 1.95 to 2.37 km in the monitoring well *OTN-2* close to the injection well *OTN-3* whereas the 4.5 kHz near-surface seismometers, sampled at 500 Hz, were
placed in wells with depths between 0.3 to 1.15 km and lateral distances of 0.6 to 8 km around the injection well (Fig. S1).

During the stimulation, the initial catalog used for evaluating the industrial success of the stimulation consisted of 6,150 events with a moment magnitude range of [-1.47 1.9], located around the injection well *OTN-3* at an epicentral distance of less than 5 km and at depth of 0.5 to 10 km (Kwiatek et al., 2019). Events with $M_W > 0.7$
were manually revised. The moment magnitudes were estimated from the industry-catalog-derived local magnitudes $M_{HEL}$ by calculating the seismic moment $M_0$ and using the formula of Hanks and Kanamori (1979) as described by Kwiatek et al. (2019). A total of 55,013 smaller events were further detected during and after the stimulation but were not located and thus not used for evaluating the stimulation success.

From the industrial catalog, we selected 3,464 events with at least 10 existing onset picks and depth
between 4.5 and 7 km. For this study, we enhanced the sub-catalog by including 321 events that occurred after shut-in of injection, i.e. after 22$^{th}$ of July 2018 at 15:52 UTC. These post-injection events have also at least 10 onset picks and moment magnitudes between $M_W$ -0.5 and $M_W$ 1.5. We manually revised 68 events of the post-stimulation seismicity with $M_W > -0.07$ and refined the P- and S-wave onset picks if necessary.

The enhanced sub-catalog including post-stimulation events was reprocessed applying a new updated
1D layered velocity model developed from P-wave onset times of calibration shots obtained during a post-injection VSP campaign (Fig. 1). Due to a low Signal-to-Noise (S/N) ratio of the VSP data, the S-wave arrival times could not be determined. Thus, the $V_P/V_S$ ratio was optimized by a trial-and-error procedure, where we ultimately constrained a $V_P/V_S$ ratio of 1.67 that minimized the cumulative residual errors of all located events, and at the same time kept the first induced events in the direct vicinity of injection point (cf. lowest injection interval in
Fig. S1).





The hypocenter locations were estimated using the Equal Differential Time (EDT) method (Zhou, 1994; Font et al., 2004; Lomax, 2005) and the new VSP-derived velocity model. In addition, station corrections were applied. The minimization of travel time residuals:

$$\left\lVert \left(T_j^{th} - T_i^{th}\right) - \left(T_j^{obs} - T_i^{obs}\right)\right\rVert_{L_2} = min, \tag{1}$$

where $T^{th}$ and $T^{obs}$ are all unique pairs (i,j) of theoretical and observed travel times of P- and S-phases, were resolved using the Simplex algorithm (Nelder and Mead, 1965; Lagarias et al., 1998) .

To further refine the quality of hypocenter locations, 2,193 events of the absolute hypocenter sub-catalog with at least 10 P-wave and 4 S-wave picks as well as hypocenter depths between 4.5 and 7 km were selected and the double-difference relocation technique (hypoDD) was applied using the new VSP-derived velocity model

(Waldhauser and Ellsworth, 2000). An iterative least-square inversion was used to minimize residuals of observed and predicted travel time differences for event pairs calculated from the existing P- and S-wave picks of the selected catalog data. The residuals were minimized in ten iterations steps. For the last iteration, the maximum threshold for travel time residuals were set to 0.08 s and the maximum distance between the catalog linked event pairs was defined as 170 m. With the hypoDD method 1,981 events were relocated and thus 90 % of the selected

2,193 events. The residuals of the relocations have a root mean square error of 9 ms. The relocation uncertainties were then assessed using a bootstrap technique (Waldhauser and Ellsworth, 2000; Efron, 1982) leading to relative location precision not exceeding ±52 m for 95 % of the catalog.

### 2.2 Spatial and temporal evolution of cumulative seismic moment

We further analyzed the spatial and temporal evolution of the cumulative seismic moment based on the relocated
seismic catalog. The cumulative seismic moment evolution with time was calculated for the entire catalog, and also separately for the three major spatial clusters. For the spatial distribution of the seismic moment, the area around the injection well was separated into horizontal bins of 50x50 m. The cumulative seismic moment of all events within each bin was then investigated by disregarding the depth.

### 2.3 Source mechanisms

To address the structural complexity of the reservoir in close proximity of the injection borehole below 4.5 km depth, source mechanisms were determined for a selected subset of events. For the 63 events with largest moment magnitudes located within the main (deepest) hypocenter cluster we first manually picked the P-wave onset polarities on the vertical component seismograms of all available stations. All waveforms were first filtered with a second order 120 Hz low-pass Butterworth filter. The same approach was applied to the 25 strongest events of





the two shallower hypocenter clusters (see Fig. 3). The focal mechanisms (FMs) were determined using the *HASH*

software (Hardebeck and Shearer, 2002). For each fault plane solution (FPS), associated uncertainties in a form of

*acceptable* solutions are provided, calculated by perturbing take-off angles and azimuths by up to 3° (95 %

confidence interval) to simulate the hypocentre location and velocity model uncertainties, respectively.

Aiming at increasing the catalog of focal mechanisms, we extended the focal mechanism calculations to

smaller events with lower S/N ratio using the cross-correlation-based technique of Shelly et al. (2016). Additional

297 small events with lower S/N ratio were processed. To this end, the waveforms from a *template* set of 70 events

with manually picked P-wave polarities were used to recover relative polarities of a *target* set of waveforms from

297 events, including 45 post-stimulation events and 18 events with manually-picked polarities. The waveforms

of the events of both sets were first pre-processed focusing on the P-wave polarities obtained from the vertical

components of all available stations. Seismograms were filtered with a second order 120 Hz low-pass Butterworth

filter and a window length of 0.064 s including 0.012 s before the P-wave first motion. After a few trials, the low-

pass Butterworth filter was fixed to 80 Hz for three stations of the satellite network due to a higher quality of the

estimated polarity results for these stations. Considering the stations separately, each extracted waveform from the

*target* set was cross-correlated with all remaining waveforms forming the *template* set. This resulted, for a

particular station and *target* event in a vector of 70 cross-correlation (CC) coefficients with the sign representing

the relative polarities between *target* and *template* P-wave onsets for a particular station. Following Shelly et al.

(2016), if the lag time of the largest cross-correlation peak was lower than 0.2 times the extracted wavelength, the

CC was accepted and used as a relative polarity estimation between *target* event and *template*. The polarity

estimates obtained from the CC values between the picked *template* and *target* events are relative and weighted

by the absolute value of corresponding cross-correlation coefficient. Thus, the sign of the estimated polarity of the

*target* event will be positive if the *template* and the *target* event have the same P-wave first motion.

To investigate the most reasonable estimated polarity pattern of each *target* event *i,* a Singular Value

Decomposition (SVD) was applied to the relative estimated polarity matrix of each station *k* to extract the strongest

signal of any *target* event obtained by the first left singular vector of the SVD (Shelly et al., 2016; Rubinstein and

Ellsworth, 2010). The estimated first left singular vectors for each station *k* are gathered in a *i*-by-*k* matrix

$$PP_{ik} = \begin{bmatrix} pp_{11} & \cdots & pp_{1k} \\ \vdots & \ldots & \vdots \\ pp_{i1} & \cdots & pp_{ik} \end{bmatrix}, \tag{2}$$

which then represents the most reasonable, however, still relative polarity pattern of each *target* event.

To reduce the polarity ambiguity of the events, we considered 18 events with known manually picked polarities

included in the *target* event set. The SVD-derived polarities of these events were compared with manually picked

polarities to investigate whether the polarities have similar or opposite signs.





Estimated polarity patterns of the events were then used to calculate focal mechanisms. For further investigation we only considered events with a good quality of estimated focal mechanisms no matter if the polarities were manually picked or estimated. Thus, we only used events with focal mechanisms that have root mean square fault plane uncertainties less or equal 35° (Hardebeck and Shearer, 2002). The final catalog of focal mechanisms included 191 events with either manually or estimated polarity pattern. The focal mechanisms generally show reverse faulting motions with NNW-SSE striking fault planes.

**2.4 Complexity of source mechanisms**

To investigate the variability of the estimated focal mechanisms, we first calculated the principal axis directions of the double-couple seismic moment tensor derived from focal mechanism for each event. To quantify the level of similarity of any two focal mechanisms, we calculated the 3D Kagan rotation angle between principal axis directions of both events (Kagan, 1991; Kagan, 2007; Tape and Tape, 2012). Low values of Kagan angle ($<20°$) suggest that focal mechanisms of two events are similar. To further group events into families with similar source mechanisms, an unsupervised classification of the 191 events was performed using a hierarchical cluster analysis based on the similarity of estimated Kagan rotation angles. Thus, the measurement of proximity *PR* of any two focal mechanisms was defined as a distance metric

$$PR_{ij} = \frac{1-\cos(\theta_{ij}^{rot})}{1.5},$$
(3)

where $\theta_{ij}^{rot}$ is a matrix containing the estimated rotation angles between any focal mechanism pair *ij*. In the following, the dendrogram tree based on the hierarchical clustering was used to separate focal mechanisms into different families.

To investigate the local stress field orientation in the reservoir surrounding the injection well, we applied the linear stress inversion method *MSATSI* (Martínez-Garzón et al., 2014) and the Bayesian-analysis-based and nonlinear stress inversion method *BRTM* of D'Auria and Massa (2015). In both methods, the strike, dip and rake angles of the fault plane solutions from the focal mechanisms were used to invert for the orientation of three stress axes. A relative measure of the stress magnitude is obtained by the stress shape ratio *R* (e.g. Hardebeck and Michael, 2006; Lund and Townend, 2007)

$$R = \frac{\sigma_1 - \sigma_2}{\sigma_1 - \sigma_3}.$$
(4)





## 3    Results

### 3.1  VSP-derived velocity model

The 1D VSP-derived velocity model shows a velocity inversion between 3 and 6 km depth (Fig. 1). The maximum
P-wave velocity is 0.15 km s$^{-1}$ larger than the maximum velocity modelled by Kwiatek et al. (2019) where a
constant velocity of 6.4 km s$^{-1}$ starting at 3 km depth was assumed. Below the velocity inversion, a constant
velocity of 6 km s$^{-1}$ is suggested from sonic logs which were used for velocity estimation between 5.1 km and
6.4 km depth. We assumed $V_P/V_S = 1.67$ considering the mean cost function uncertainties of the absolute
hypocenter locations for different $V_P/V_S$ ratios as well as the spatial distribution of the initial events around the
open hole of the injection well. This is slightly lower than the $V_P/V_S$ ratio of 1.68 used for the velocity model
presented in Kwiatek et al. (2019).

### 3.2  Seismic catalog update

We extended the original seismic catalog analyzed in Kwiatek et al. (2019) by 321 events that occurred after the
stimulation campaign. In total 3,785 events were located in absolute sense using the new VSP-derived velocity
model and refined P- and S-wave picks. We further relocated 1,981 events with at least 10 P- and 4 S-wave picks
applying the double-difference relocation method. The expanded event catalog together with the event detection
is available as data publication (see section *data availability*).

The selected sub-catalog used for absolute hypocenter locations consists of 3,785 events with magnitudes
between $M_W$ -0.8 and $M_W$ 1.9. The moment magnitudes of the absolute located seismicity is plotted with time
during and after shut-in in Fig. S2. The five different stimulation phases (P1-P5) performed in 2018 are also shown
in Fig. S2 in combination with the wellhead pressure. Further details of the stimulation protocol and seismicity
evolution are presented by Kwiatek et al. (2019), and here we focus on analysis of post-stimulation seismicity.

The 321 post-injection events were detected during a time period of two month after shut-in of injection
and displayed magnitudes $M_W \geq$ -0.5. After shut-in, the seismic event rate increased shortly and started to rapidly
decrease after bleed-off of the well (Fig. 2). This decrease in activity continued until the 5$^{th}$ day after the end of
the injection followed by a slower decrease thereafter. During the first two days after shut-in, seven events with
$M_W \geq 1.0$ occurred. The largest event had a magnitude of $M_W = 1.5$ and occurred directly after bleed-off, followed
closely by two $M_W$ 1.3 events. Two events with $M_W \geq 0.9$ occurred within the first 11 days of the post-stimulation
phase. Two further $M_W > 1$ events occurred within 24 hours and 17 days after the stimulation ended, one with
moment magnitude of 1.6 (Fig. 2). The latter events coincided with engineering operations performed in the
injection well.





The updated relocated hypocenters of 1,981 events with at least 10 P-wave and 4 S-wave picks and magnitudes between $M_W$ -0.7 and $M_W$ 1.9 occurred in three spatially separated clusters elongated in southeast (SE) - northwest (NW) direction and centered along the injection well in good agreement with Kwiatek et al. (2019) (Fig. 3). Elongation of the clusters in SE-NW direction is sub-parallel to the local maximum horizontal stress $S_H^{max}$ = 110° (Kwiatek et al., 2019; Heidbach et al., 2016; Kakkuri and Chen, 1992). The main seismicity cluster centers around the open-hole section of the borehole. The uppermost hypocenter cluster is spatially separated into one main cloud and a second smaller cloud (Fig. 3b). The events within the smaller cloud mainly occurred during the two last stimulation phases (P4-P5) and thus, the separation is also recognizable in time domain. The main cloud of the uppermost cluster spans about ~300 m in depth separated ~100 m from a smaller cloud with ~150 m vertical extend. The deepest hypocenter cluster spans ~700 m depth. This exceeds vertical relocation precision, which is well constrained due to sensors located in a vertical borehole. The spatio-temporal seismicity evolution during the stimulation developed in two preferential directions starting from the injection well: in NW-SE direction sub-parallel to the direction of $S_H^{max}$ as well as in northeast (NE) direction with depth.

68 post-stimulation events with at least 10 P- and 4 S-wave onset picks could be relocated using the double-difference technique and are shown as grey dots in Fig. 3. The post-stimulation events are mainly located at the outer edges of the clusters following the trend observed during the stimulation. The post-injection seismicity shows no spatial migration and seems to be mostly confined to three isolated clusters, with two of them located on the NW flank of the injection well *OTN-3* (Fig. 3a). The largest post-stimulation events with magnitudes between $M_W$ 1.0 and $M_W$ 1.5 occurred at the NNW and SSE outer edge of the main cluster. These events are located in close proximity to some of the largest events of the last stimulation phase P5 (red rectangles in Fig. 3), when high seismicity rates were observed.

### 3.3 Temporal evolution of cumulative seismic moment

For the stimulation period, the temporal evolution of the cumulative seismic moment release is discussed by Kwiatek et al. (2019). Here, we show the temporal evolution of the cumulative seismic moment ($CM_0$) release during post-stimulation period and compare it with the evolution before shut-in of injection. During the first two days of the post-stimulation period, the increase of $CM_0$ was similar to the first two days of stimulation phases P1-P5 (Fig. 4). Shortly after bleed-off, the $CM_0$ rapidly increased due to the three $M_W \geq 1$ events (Fig. 2). Thereafter, the increase of post-stimulation moment release was substantially less compared to a similar time period during P1-P5. Only two single events occurred with $M_W \geq 1$ during day 17, seemingly triggered by post-stimulation engineering operations in the well.





The temporal evolution of the $CM_0$ separated for each hypocenter cluster is shown in Fig. 5. For the upper cluster, the increase in the $CM_0$ is visibly larger for the stimulation phase P1 than for the other phases. For stimulation phase P2, a substantial increase in $CM_0$ occurred between day 4 and 5. For the central hypocenter

cluster, a substantial increase in the $CM_0$ is visible for stimulation phase P2, P4 and P5 at the beginning of day 3 and also for P1 and P4 during day 6. For both upper and central clusters, the post-stimulation $CM_0$ is substantially smaller compared to that from injection (Fig. 5a-b). The $CM_0$ during post-stimulation in bottom cluster is similar to P2-P5 within the first two days and afterwards lower than P2-P5 for the main cluster. Inevitably, the bottom cluster that hosts the majority of the seismic activity also display the highest $CM_0$ (Fig. S3). We note that the slopes

of the $CM_0$ evolution are similar for the upper and central cluster, but steeper for the bottom cluster (Fig. S3).

### 3.4 Spatial evolution of cumulative seismic moment

During stimulation, the largest moment release and level of seismic activity occurred at the center of the main event cluster at the bottom of the injection well close to the open-hole section (Fig. 6a-b). Furthermore, larger events in the main cluster tend to locate at the greatest depths. Interestingly, a NNW-SSE alignment of enhanced

cumulative seismic moment release is visible in the main hypocenter cluster in agreement with the preferred NW-SE trending direction of the two upper hypocenter clusters. The hypocenters of larger events show a similar alignment (Fig. 6a, S4). A smaller area at the NNW outer flank of the bottom hypocenter cluster displays anomalously high $CM_0$ release caused by large events occurring during the last injection phases and after injection (red rectangle in Fig. 6a-b). Interestingly, epicenters of two tectonic seismic events with $M_W$ 1.4 and $M_W$ 1.7 were

reported to occur in 2013 a few kilometers NW of the bottom hole section of well *OTN-3* (Kwiatek et al., 2019).

### 3.5 Complexity of source mechanisms

We determined 191 single-event focal mechanisms (Fig. 7). Using the dendrogram tree based on hierarchical clustering (Fig. S5), events were separated into three distinct families (I-III) with similar focal mechanism orientations containing 99, 60 and 27 events, respectively (different coloring of beach balls in Fig. 7). Five events

were not grouped in any of the three families and thus, were not considered any further. Events belonging to the three families are not separated spatially. Oblique reverse faulting is the dominant source mechanism type, which is in contrast to the regional strike slip regime (Kwiatek et al., 2019). The two largest events with reverse faulting were classified into family III. Fault plane solutions from all families indicate a range of preferred SSE-NNW to SW-NE strike directions, sharing comparable dips ranging approx. 35-50° (Fig.7a and 7e). The source mechanisms

of only a few events indicate strike-slip faulting, with two of them occurring after shut-in. A total of 14 estimated



focal mechanisms are post-stimulation events (Fig. 7b, 7d and 7f). The post-stimulation events contained in the main hypocenter cluster at the bottom of the well have similar focal mechanisms as events during the stimulation. In the central hypocenter cluster, two strike-slip events occurred close by.

To further explore separation of the focal mechanisms into distinct families, we analyzed the rotation angle between principal P- and T-axes as a measure of mechanism (dis)similarity. We first calculated mean fault plane solution for each family. The strike/dip/rake-values of the mean fault plane solutions (FPS) for family I, II and III are 332°/47°/43° and 32°/51°/141° and 67°/36°/122°, respectively. The focal mechanisms with mean fault plane solutions and all best FPSs of each family are plotted in Fig. 8a-c. Hillers et al. (2020) recently estimated focal mechanisms for the 14 largest events for which the majority is similar to family I FMs. The calculated

rotation angles between mean solutions of family I and II, I and III, II and III are 71°, 59° and 53°, respectively. Taking into account that focal mechanisms are assumed to be similar if the Kagan rotation angle is less than 20°, none of the three families is similar to each other. Difference between family I and II is the most prominent, whereas rotations I-III and II-III are comparable. However, despite mean solutions of different families are quantitatively distinct, the individual mechanisms are not necessarily very different (Fig. 8d-f) in between families.

The total P-axis uncertainties are strongly overlapping between three families. At the same time, the T-axes uncertainties form three distributions that, while compared between families, are only partially overlapping. This overall suggest that the FPSs may be sensitive to changes in polarities on individual stations located close to the nodal plane.

     In the following, we analyzed qualitatively the polarity patterns of events forming three families. The

most repetitive polarity pattern observed at each station for a particular family is plotted in Fig. 8a-c. We first verified consistency of polarity patterns for events with manually picked polarities (N=37/15/15 FPSs for family I, II, III, respectively). We noted the strike slip mechanisms are attributed to least well-constrained focal mechanisms belonging to family II. The main substantial difference in the polarity patterns across families seems to be related to polarities observed at two stations *MALM* and *MUNK* (Fig. 8a-c). For family I, the polarities on

these two stations are positive and extremely consistent among events forming the family (35 out of 37 events display such a behavior). For family II, we observe *MALM* and *MUNK* to have mostly negative and positive polarity pattern, respectively. For family III, the situation is reversed with *MALM* and *MUNK* having predominantly positive and negative polarity pattern, respectively. We further analyzed qualitatively the polarity pattern of events with polarities estimated from cross-correlation based technique of Shelly et al. (2016). Here, the

situation generally further complicates due to appearing ambiguities in resolving the polarities due to decreased signal-to-noise ratio. However, for the majority of the events forming family I, the resolved focal mechanisms still show a consistent polarity pattern to that from manually picked ones, with only incidentally changing polarities





on stations *UNIV* and *RUSK* located away and thus displaying lower signal-to-noise ratio. The pattern of resolved polarities for family II is generally comparable to that resolved for manual polarities. However, 19 out of 45 events

have negative estimated polarities for *MALM* and *MUNK*, thus the resolved polarity patterns seem to vary more in comparison to that of family I. The events with estimated polarities for family III have the same patterns for stations *MALM* and *MUNK* as the manually picked events except of one event. However, other stations with lower signal-to-noise ratio display sometimes varying resolved polarities. We suppose that 1) the attribution of focal mechanism to a particular family is substantially depending on polarity pattern of limited number of stations that are being

close to the nodal planes, and 2) family I focal mechanisms seem the most stable.

Using the *BRTM* and *MSATSI* stress tensor inversion methods based on 191 focal mechanisms, we estimated the local stress field orientation. The variability of FMs to constrain the stress field inversion is given due to high Kagan rotation angles between the mean FPSs of the three families with 53° to 71°. The *BRTM* results show that the maximum principal stress axis $\sigma_1$ is oriented almost horizontally with a trend of 279° and a plunge

of 4° (Fig. 9). The minimum principal stress axis $\sigma_3$ has a trend and plunge of 185° and 67°, respectively. The stress shape ratio is calculated with $R = 0.53$. The estimated orientation of $\sigma_1$ deviates ~10° from the local maximum horizontal stress $S_H^{max}$ (Kwiatek et al., 2019). Using the *MSATSI* method, the trend and plunge of $\sigma_1$ is calculated with 271° and 11°, respectively. Thus, the estimated trend of $\sigma_1$ deviates ~20° from the maximum horizontal stress $S_H^{max}$. The minimum principal stress axis $\sigma_3$ is oriented with a trend of 76° and a plunge of 79°.

The stress shape ratio is with $R = 0.72$ larger compared with the *BRTM* estimate.

The stress inversion of the induced seismic events represents a local reverse faulting regime. This is in contrast to the regional strike-slip regime estimated from regional stress and borehole data (Kwiatek et al., 2019). Only the focal mechanisms of a few events present a dominant strike-slip faulting, which typically are smaller events with a less well constrained polarity pattern.

**4    Discussion**

Analysis of the seismic data suggests that fluid injection was performed into a complex network of small-scale pre-existing and distributed fractures and minor faults, rather than activating a single, major fault (Kwiatek et al., 2019). In an effort to characterize the structural complexity of the reservoir in detail, we compiled a high-resolution dataset of hypocenters and single-event focal mechanisms by enhancing and refining the original seismic catalog.

The relocated events of our updated catalog show three separated spatial hypocenter clusters along the injection well in good agreement with Kwiatek et al. (2019) and Hillers et al. (2020). Hillers et al. (2020) used seismic data collected from an independent surface-based seismic network of dense sub-arrays, whereas Kwiatek





et al. (2019) used the same seismic network as we do but a simplified velocity model and slightly different $V_P/V_S$ ratio. The hypocentral depths of the events vary slightly between this and previous studies. We found that differences between absolute locations among these catalogs are likely explained by variations in $V_P/V_S$ ratios and velocity models.

We also provide the first analysis of post-stimulation events expanding the seismic catalog to investigate potential changes in the seismicity pattern from stimulation to the post-stimulation period. Compared to the seismicity occurring during the stimulation, the post-stimulation seismicity shows no spatio-temporal migration and remains largely confined to three separate clusters. One cluster arose after bleed-off and is located at the NW flank of the central hypocenter cluster that formed during stimulation. The largest post-stimulation events occurred at the NNW and SSE outer edges of the main hypocenter cluster where also anomalously higher seismicity rate and larger events were observed during the last stimulation phase P5 (cf. Fig. 3). For the main hypocenter cluster, the temporal evolution of the post-stimulation $CM_0$ shows similarities to the injection period until bleed-off of the well with only small changes thereafter. This suggests that seismicity is driven by the elevated pressure in the reservoir due to the previous hydraulic pumping (=increased stored elastic energy). However, hypocenter propagation requires active pumping. This is indicated by a much smaller residual increase in $CM_0$ and no further migration of the seismicity after bleed-off and decrease in reservoir fluid pressure.

The spatio-temporal seismicity evolution during stimulation as well as the spatial distribution of the cumulative seismic moment release indicate clear alignment of the events in NW-SE direction in the two shallower hypocenter clusters which could signify activation of permeable zones along faults or joints oriented in this direction. Existence of these zones is supported by the results of *OTN-3* well logging, where intervals of highly damaged rocks were detected that roughly coincide with the intersection of the upper seismicity clusters and the well path. For the largest bottom seismicity cluster, the relocated seismicity is distributed diffusively around the injection well. However, larger seismic events form a distinct alignment along a NNW-SSE direction (Fig. 6a, S4) with post-stimulation events clearly located at the perimeter of the narrow zone (Fig. S4). This alignment indicates activation of another permeable zone similar to the two upper ones. The NNW-SSE trending orientation is coinciding with abundance of very similar focal mechanisms from the best constrained family I events with strike direction nearly identical to the NNW-SSE alignment of hypocenters. Moreover, two natural micro-earthquakes with $M_W$ 1.7 and $M_W$ 1.4 occurred in 2013 a few kilometers NNW from the well (Kwiatek et al., 2019). Although there is no detailed information available on their depths due to limited coverage of the seismic network at their origin time, their epicentral location coincides with the NNW perimeter of the bottom NNW-SSE alignment hosting large induced seismicity events as well. These observations suggest that the stimulation activated at least three prominent NW-SE to NNW-SSE oriented permeable zones of subparallel fractures or faults that are



responsible for seismicity migration away from the injection well during the stimulation. The deepest NNW-SSE
         trending zone is buried in a more disperse seismic activity forming the bottom cluster and hosts the largest induced
         (and likely earlier some natural) earthquakes. The fact that the largest events occurred in the bottom permeable
         zone may be simply related to the highest expected pore pressure perturbation in this volume due to injection and
         migration of fluids. Kwiatek et al. (2019) speculated that the maximum event magnitude is either limited by
available fault sizes or strength of the faults. The total length of NNW-SSE trending permeable bottom zone
         (~650 m, Fig. S4), clearly marked by the numerous and very similar focal mechanisms, is much larger than the
         average size of a single $M_W$ 2 earthquake (~80 m diameter) with even lower relocation precision. We therefore
         suggest that the upper limit to maximum magnitude is related to the low fault strength.

         For the main hypocenter cluster, the seismicity migrates progressively beyond the injection intervals
towards NE and towards greater depths, dipping in the same direction as the inclined portion of *OTN-3* well
         (Fig. 3). The depth propagation of the seismicity may be affected by gravity of the cool water into warm and less
         dense pore fluid of the reservoir as e.g. observed at The Geysers geothermal field (Kwiatek et al., 2015). The
         downward propagation of seismicity may signify activation of small-scale fractures striking NNW-SSE and
         dipping along the injection well. This is again supported by the catalog of source mechanisms forming family I
events (cf. Fig. 7 and 8a). To further understand this striking observational and qualitative agreement of family I
         fault planes with spatial distribution and evolution of seismic activity, we tested which family of focal mechanisms
         is better oriented for failure within the local stress field. A projection of estimated FPSs in a Mohr circle diagram
         reveals fault plane orientation with respect to the stress field (Fig. 10). Optimally oriented fault planes are more
         likely to be activated (e.g. Vavryčuk, 2011), especially for weak faults. To calculate the failure criterion, we
assumed a friction coefficient of $\mu = 0.7$ as a mean value for faults in the Earth's crust (Vavryčuk, 2011). While
         projecting the selected one of the two nodal planes from each fault plane solution, we used the nodal plane that
         displayed higher instability coefficient $I$ (cf. Vavryčuk, 2014; Martínez-Garzón et al., 2016):

$$I = \frac{\tau + \mu(\sigma_n + 1)}{\mu + \sqrt{1 + \mu^2}},\tag{5}$$

         with $\tau$ and $\sigma_n$ as the normalized shear and normal tractions, respectively and $\mu$ as the friction coefficient.
Clearly, FPSs from family I are the most favorably oriented with respect to the local stress field (blue
         points and triangles in Fig. 10), as also indicated by the highest fault instability coefficients (Fig. S6). It turned
         out that the most optimally oriented fault plane is always the one trending NNW-SSE and dipping approximately in
         the direction of inclined portion of *OTN-3* well (indicated by P1 nodal planes in Fig. 8a). This is also confirmed
         by the mean solution of family I (332°/47° plane, blue P1 marker in Fig. 10) displaying the highest instability
(Tab. S1). However, also the fault planes represented by the auxiliary plane of the mean solution of family I are



quite favorably oriented (blue P2 marker in Fig. 10). Some of the family III events are also quite favorably oriented with the stress field. We note that instabilities of auxiliary planes of mean FPSs for family I and III are similar (green and blue P2 dots in Fig. 10, Tab. S1), in agreement with their mean auxiliary nodal plane orientations of 210°/60° (P2 in Fig. 8b-c). Qualitatively, nodal planes from family II seem to be mostly unfavorably oriented with

the stress field (orange points and triangles in Fig. 10), as indicated by the lowest instability coefficients (Fig. S6). However, some P1 nodal planes are striking N-S (cf. Fig. 8b) and thus showing quite similar orientations as the P1 FPSs of family I (Fig. 8a), leading to higher instability coefficients for these planes (orange dots and triangles close to blue and green P2 marker in Fig. 10). Here, we found 19 events of family II show in fact similar polarity patterns than that observed for family I events with only an opposite polarity for station *MUNK*.

The performed analysis of fault instability clearly showed that high-quality focal mechanisms constituting family I events display comparable oblique reverse component and optimally oriented fault planes striking approximately NNW-SSE and dipping around 45°. These fault plane orientations are in agreement with the estimated stress field, and they explain well the spatio-temporal evolution of seismicity with corresponding fluid migration pattern. The 2018 seismic activity lightened up a pre-existing network of small-scale parallel fractures

dipping to ENE, in agreement with the dip direction of the inclined part of the injection well. Fault planes striking NNE-SSW to NE-SW and dipping around 60° were also indicated to be quite favorably oriented with the stress field represented by the auxiliary plane of the mean FPSs for family I and III. Drill bit seismic data suggest the existence of a steeply dipping NE-SW striking structure which might be activated by the 2018 seismic activity. We note the FM results are in good agreement with a limited number of 14 focal mechanisms of the strongest

events presented in Hillers et al. (2020), which were all but one displaying reverse faulting motions.

## 5    Summary and conclusions

We present a new seismic catalog for the geothermal stimulation in Helsinki 2018 determining new locations and relocations on the basis of the new VSP-based velocity model and include the post-stimulation seismicity resulting in a catalog with 3,785 events. The catalog is extended by the list of detections, accounting to 61,163 events

provided to scientific community. The magnitude of completeness of the entire catalog is $M_C = -1.0$. The catalog is supplemented by 191 focal mechanisms calculated using polarity-based and cross-correlation based methods and is used to discuss the structural complexity of the reservoir.

Spatial migration of the seismicity is driven by enhanced pore fluid pressure due to active injection, as no spatial migration of the post-stimulation seismicity after bleed-off is found. The temporal behavior of the post-

stimulation seismic moment release until bleed-off is still similar to the moment release observed during individual
stimulation phases.

An activated network of at least three NW-SE to NNW-SSE oriented fracture zones of up to 200 m
thickness seems to be responsible for the significant seismic activity migration towards NW-NNW and SE-SSE
away from the injection well. The deepest fracture zone also hosts much of the larger seismic events with
magnitudes exceeding $M_W \geq 1$, suggesting elevated fluid volume and pore fluid pressure, leading to accumulation
of hydraulic energy in this area, relaxed in larger seismic events.

Best-constrained focal mechanisms strike NNW-SSE in agreement with orientation of three fracture
zones. Most of these mechanisms display ~45° ENE dipping oblique-thrust fault planes that were found to be
critically stressed in the resolved local stress field. These fault kinematics explains well NNW-SSE migration of
seismicity along damage zones, as well as the gravitation-driven downwards migration of smaller events towards
NE-NNE, along the dip direction vector of the inclined portion on injection well.

We conclude that seismic slip occurs on sub-parallel network of favorably oriented pre-existing fractures,
but weak faults striking in NNW-SSE direction and dipping 45° ENE. The localization of seismic moment release
in NNW-SSE trending zones suggest existence of NNW-SSE trending damage structures or lithological
differences that increase the mobility of fluids in this confined parts of the reservoir.

**Data availability**

The seismic event catalog will be available through GFZ data services: http://dataservices.gfz-potsdam.de/portal/.
For the event detections, the catalog contains origin times, local and moment magnitudes. For located events, the
catalog contains origin times, local as well as moment magnitudes, absolute locations in local Cartesian coordinate
system and for relocated events also the double-difference relocated locations in local Cartesian coordinate system.

**Competing interests**

The authors declare that they have no competing interests.

**Author contribution**

M.L.: data reduction, analysis and results interpretation, and draft version of the manuscript. G.K. and P.M.-G.:
data analysis, results interpretation, and manuscript correction. M.B., G.D, and P.H.: results interpretation and





manuscript correction. T.S.: project management, drilling and stimulation program development and managing, and manuscript correction.

## Acknowledgments

We thank Ilmo Kukkonen and Peter Malin for the valuable discussions. G. K. acknowledges founding from DFG
(German Science Foundation), Grant KW84/4-1. P. M.-G. acknowledges funding from the Helmholtz Association through the Helmholtz Young Investigators Group "Seismic and Aseismic Deformation in the brittle crust: implications for Anthropogenic and Natural hazard" (http://www.saidan.org).

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





**Figures**

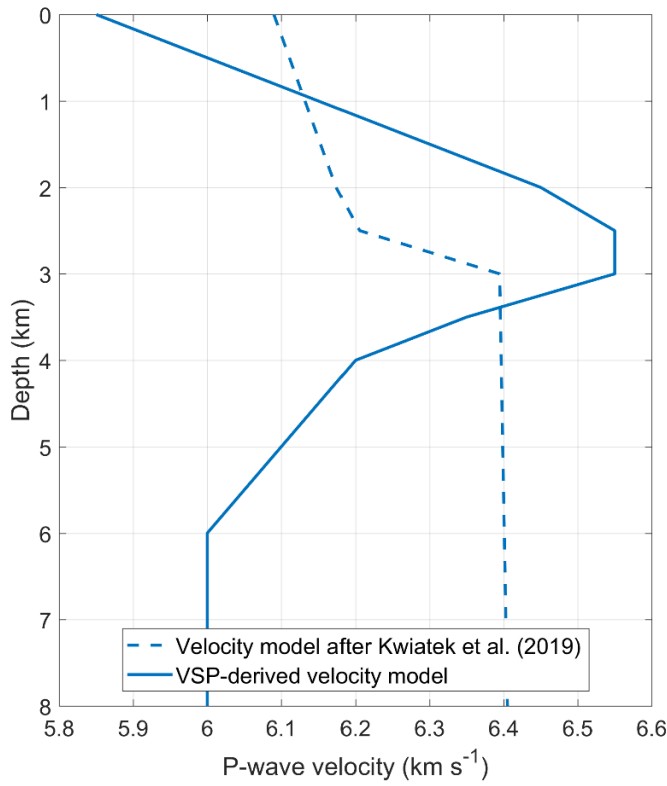

**Figure 1. Comparison of the updated 1D layered velocity model derived from calibration shots of VSP campaign (solid**
**line) with the 1D layered velocity model used in Kwiatek et al. (2019) (dashed line).**





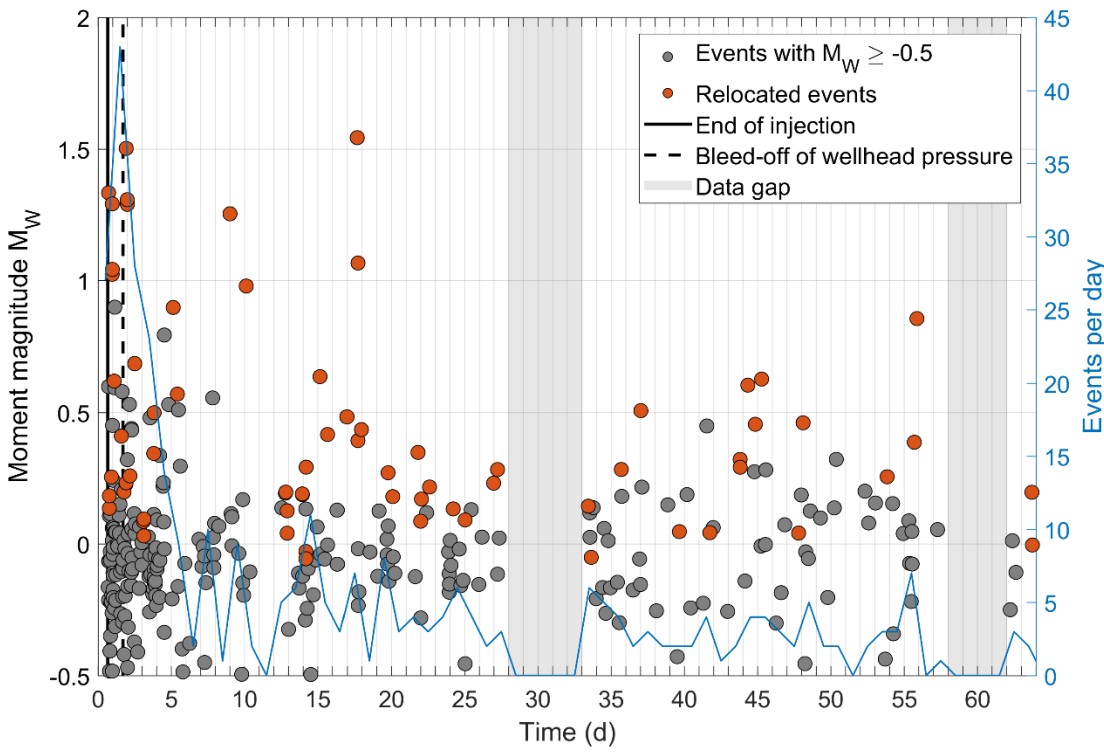

**Figure 2. Post-stimulation seismicity plotted with time. Events with $M_W \geq$ -0.5 and relocated events are plotted as grey and orange dots, respectively.**




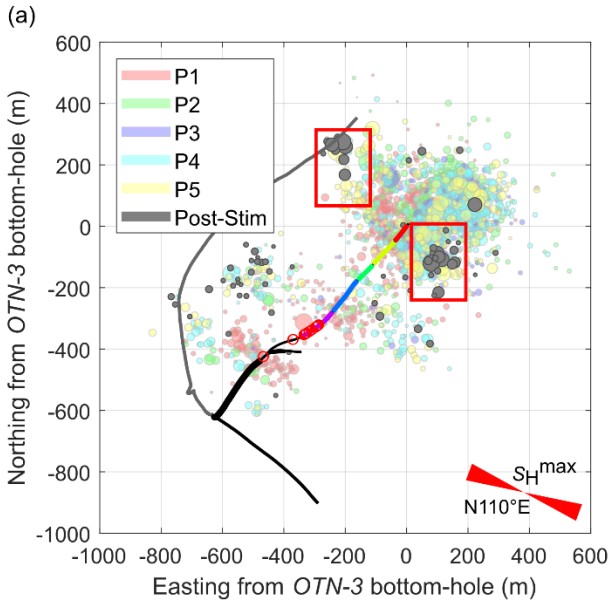

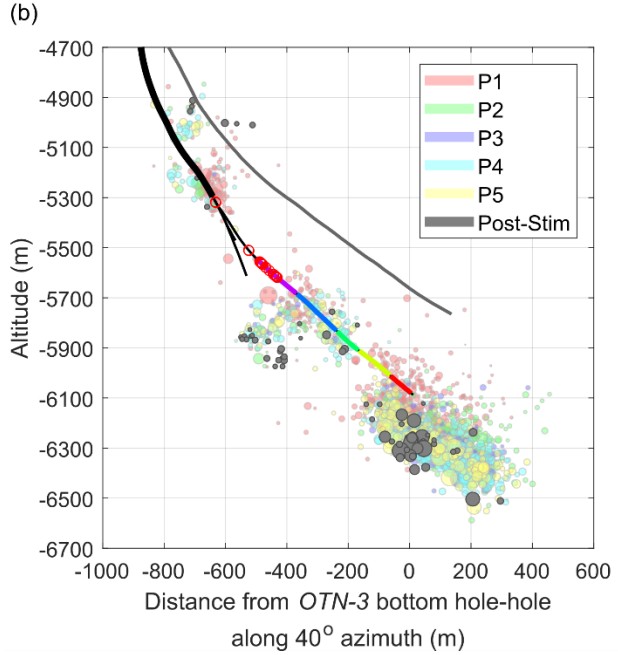

**Figure 3. Hypocenters of relocated events. (a) Map view and (b) SW-NE depth section. The hypocenters are color-coded with the stimulation phases (cf. Kwiatek et al., 2019) and size corresponds to moment magnitude. Relocated seismicity that occurred after the stimulation is represented as grey dots. Areas with large events occurring during stimulation phase P5 and post-stimulation time are highlighted by red rectangles (see main text for details). The new *OTN-2* well (grey) was drilled in 2019 to 2020 after the stimulation.**




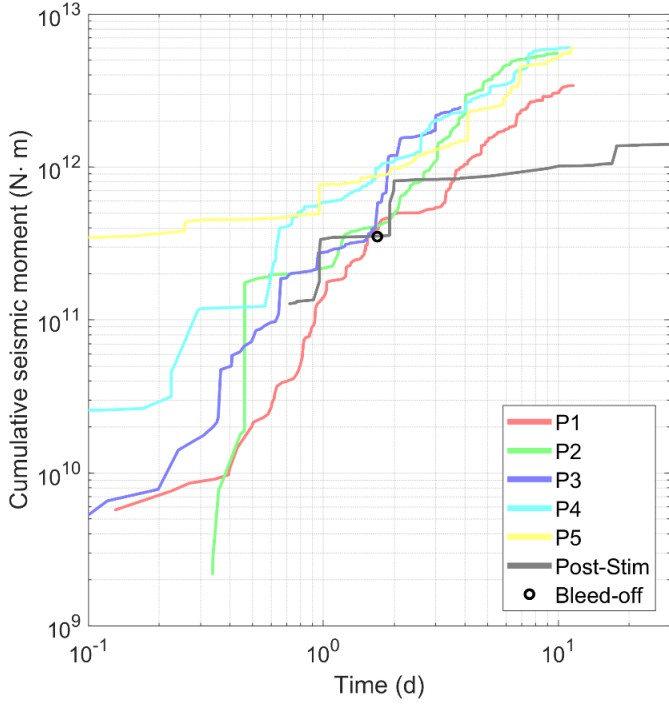

**Figure 4. Temporal evolution of cumulative seismic moment release for the relocated seismicity since the beginning for each injection phase as well as for the post-stimulation phase.**


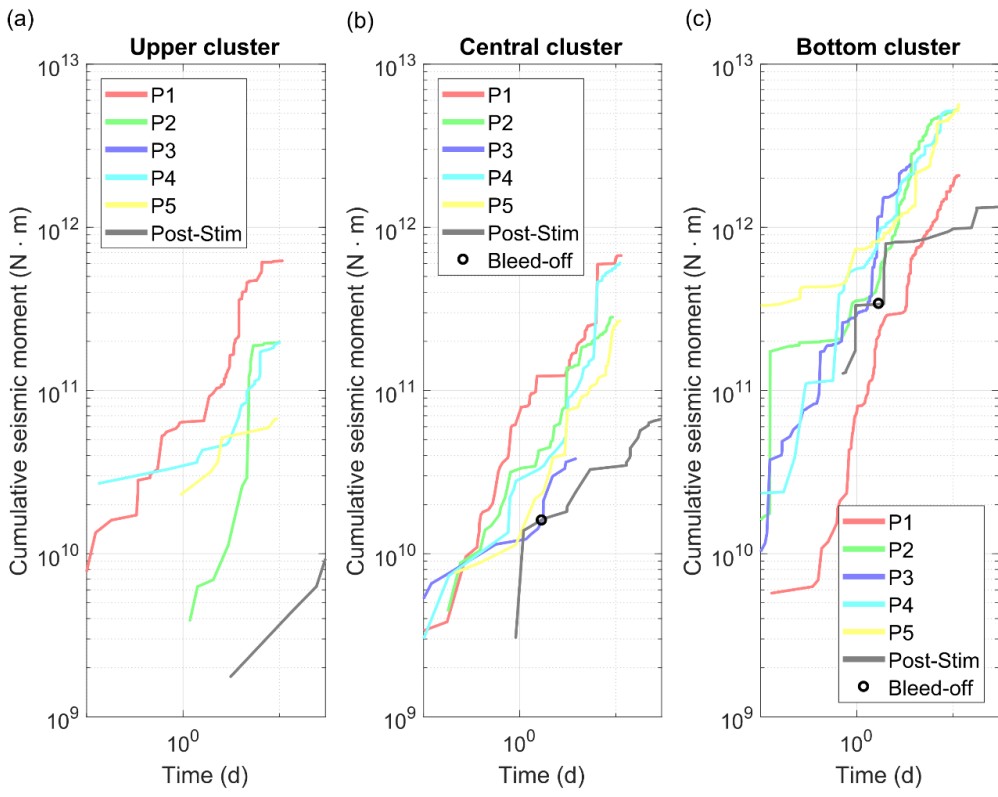

**Figure 5. Temporal evolution of the cumulative seismic moment release with time for each of the three hypocenter clusters separately: (a) The uppermost hypocenter cluster, (b) the central hypocenter cluster and (c) the deepest and main hypocenter cluster.**






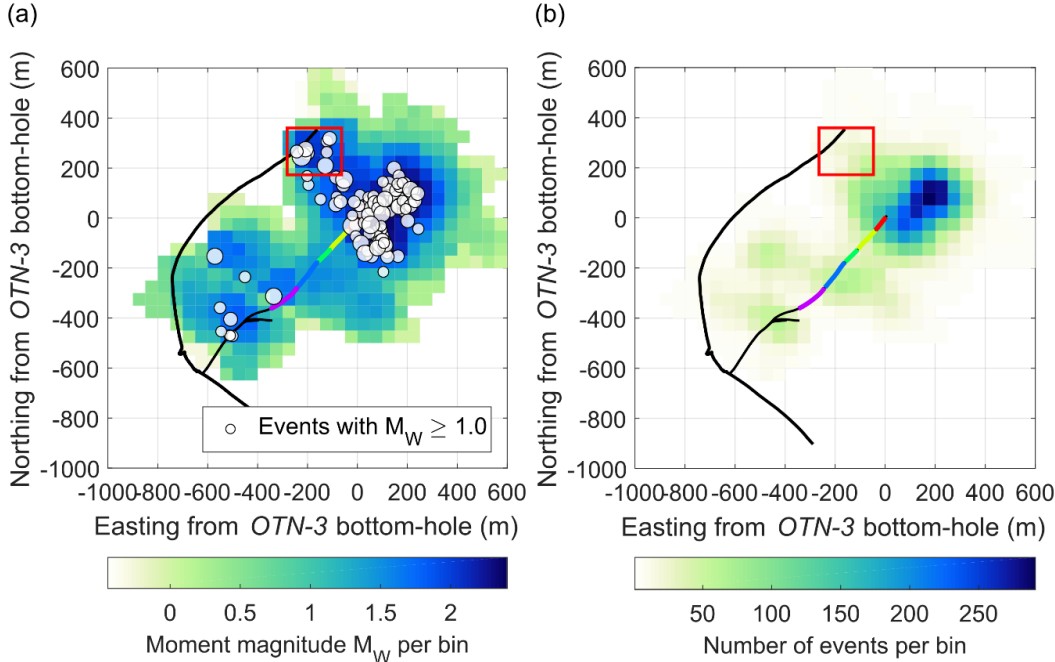

**Figure 6. Spatial evolution of the cumulative seismic moment release of the relocated seismicity per bins of 50-by-50 m.**
**(a) The cumulative seismic moment release converted to seismic moment magnitude per bin overlaid by seismicity with**
$M_W \geq 1$. **(b) The number of events that occurred per bin. A smaller area of anomalously high $CM_0$ release caused by a**
**few large events is highlighted by red rectangle.**





**Figure 7. Orthogonal views of estimated focal mechanisms in three different projections: (a, b) map view, (c, d) side view from south (180°) as well as (e, f) side view from NW (290°), along the direction of the maximum horizontal stress $S_H^{max} = 110°$. (a, c, e): All 191 estimated focal mechanisms. (b, d, f): Focal mechanisms of post-stimulation events. Color-code indicating family obtained. Relocated seismicity without estimated focal mechanisms are plotted with grey small dots.**



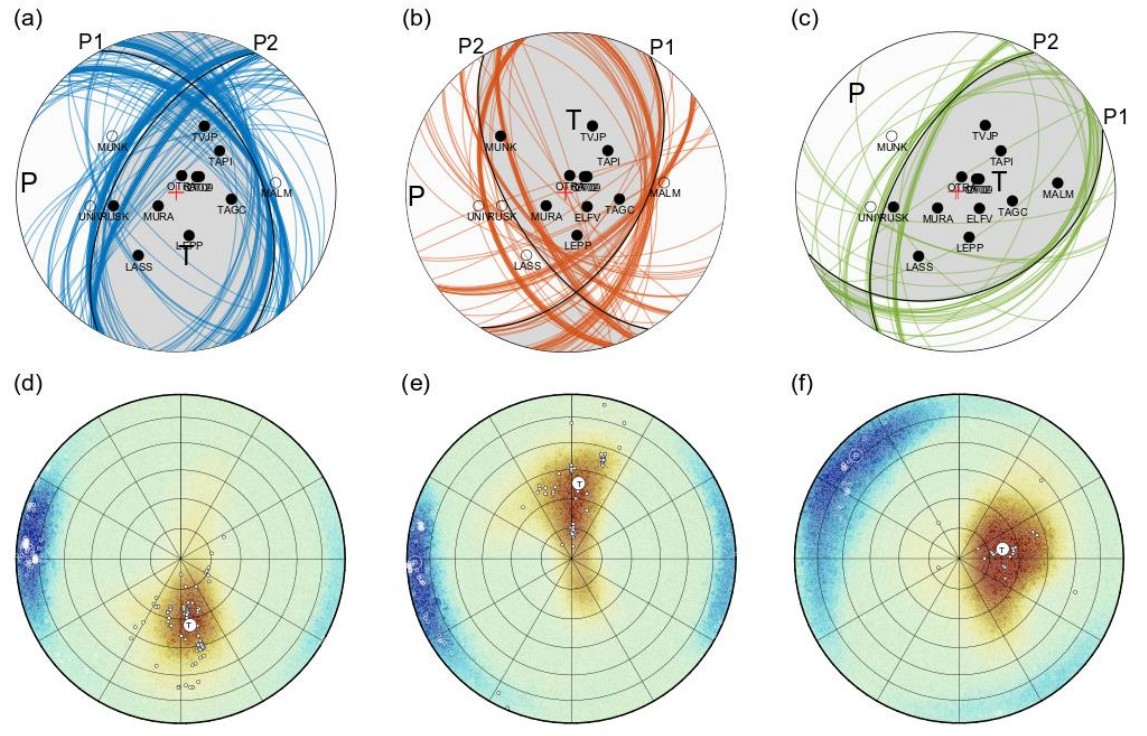

**Figure 8. (a-c): Mean fault plane solutions (black lines) calculated from best FPSs of events forming family I (a), family II (b) and family III (c). Contributing FPSs from which mean is calculated are shown with blue, orange and**
**green color, respectively. The most repetitive polarity pattern observed at each station is presented as black or white dot for positive or negative onsets, respectively. P1 and P2 symbols correspond to the projections of main and auxiliary fault planes according to which one is better oriented for failure on the Mohr circle represented in Fig. 10. (d-f): For each of the families, the mean P- and T-axes as well as axes of contributing FPSs are plotted with big and small white dots, respectively. The *HASH*-derived uncertainties (95 % confidence interval) of the P- and T-axis of all events within**
**each family are shown using blue and brown coloring scale, respectively.**





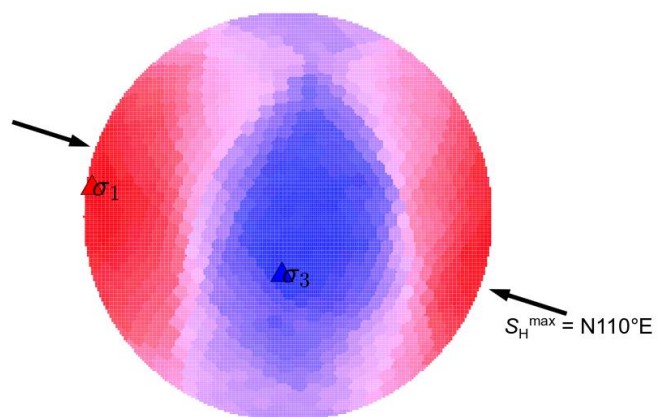

**Figure 9. Stereonet of the estimated local stress field using *BRTM* method. Red and blue triangle represent maximum and minimum principal stress axes σ₁ and σ₃, respectively. Black arrows represent maximum horizontal stress $S_H^{max}$ in the reservoir.**

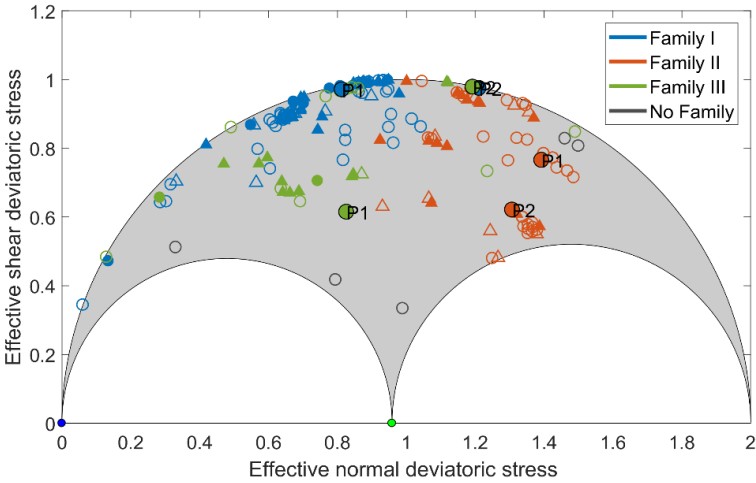

**Figure 10. Deviatoric Mohr circle representing the local stress field with the fault plane solutions having the highest fault instability coefficient of the estimated focal mechanisms. Events with $M_W \geq 1$ and $M_W < 1$ are plotted as triangles and circles, respectively. Filled and unfilled markers represent events with manually picked and estimated polarities, respectively. The mean and its auxiliary fault plane solution of each family are plotted as filled large dots labelled with P1 and P2, respectively. Most family I events (blue symbols) occurred on critically stressed faults.**