# Peer review of "Seismicity during and after stimulation of a 6.1 km deep Enhanced Geothermal System in Helsinki, Finland"

_Solid Earth, 2020_

## Referee Comment (RC1) · Anonymous Referee #1 · 28 Oct 2020

The manuscript "Seismicity during and after stimulation of a 6.1 km deep Enhanced Geothermal System in Helsinki, Finland" brings an extended results of processing the seismic monitoring data set obtained during the hydraulic stimulation carried out in 2018. Data from different available seismic stations were combined to extend the number of detected and located events and to display the seismic moment release in time. Cross correlation technique was used to estimate focal mechanisms of the largest possible number of events whose variability was analyzed and used to determine the stress field components. The instability of fault planes was then used to assess the most prominent faults.

[Figure]

The study deals with very interesting data on injection induced seismicity in a unique experiment and gives some valuable results. These are in particular the extended catalog, the focal mechanisms and principal stresses. Providing these data to the scientific community will undoubtly help better understanding the induced seismicity in geothermal projects in hard rocks. However, despite of reasonable language (as I can assess as non-native speaker), the study is not easy to read. This holds e.g. to the parts on catalog methodology and results, which is not easy to understand. One of the reasons is structuring the paper to Methodology and Results sections. It is a good approach in general, but in some cases it breaks the individual topics and makes the paper longer and understanding more difficult. So I recommend to describe only the more sophisticated methods like 2.3, 2.3 and the location part of 2.1

I also think that the spectrum of methods applied is too wide with no clear focus. The authors should decide if they present new high quality extensive seismic catalog whose parameters are characterized by a set of suitable (statistical) methods or they present a seismological study including interpretations. The point is that despite the catalog is the most valuable output, it is never characterized by at least Gutenberrg-Richter distribution and similar methods. The authors also spent a lot of effort determining focal mechanisms using quite sophisticated method to get maximum number of mechanisms, they however do not show the whole set of FM and assess their quality. I am also not sure about the improved quality of locations in terms of their asymmetric position to the borehole. As a result I believe the paper should be restructured according to its main focus - presentattion of new data. Details of my comments which should be adressed in a major revision are summarized below.

Particular comments

Ln 109-120 (Methodology). The explanation about different subsets of larger and smaller events and their relocation is not very clear. E.g. how many events were above Mw 0.7; were the 3464 events chosen from this subset?; did these events occur during stimulation because you added 321 post-stim events?; did 68 events com from

this subset?. . .

Ln 172 - 176. Please explain the SVD application in more detail. The point is that SVD is usually used to find a common pattern in a data set. For this you would need more polarity patterns for each event that just one, which you have as a result of cross correlation.
The next question is whether the polarity matrix (eq. 2) shows the polarity fit between the target and template events as indicated on Ln 171 or the fit of polarities themselves. In the first case, it could not be used for calculating focal mechanisms.

Ln 178. The way you reduced the polarity ambiguity is not clear; by considering manually picked events one can verify the automatic picks, I believe.

Ln 185 The final sentence mentioning the resulting reverse faulting fits rather to the Results than Methodology section

Ln 195 Please argue for using this distance metrics - what is the reason for 1.5 in the denominator? And which type of cluster analysis did you use? What is the difference to the published method of moment tensor clustering of Cesca (2014)?

Ln 209-216 (Results) I think that the VSP based model deserves more attention. The present way is not appropriate - to show the model as a result without any more details. If it is considered as a result of this study, the data, methods and results should be shown. In the opposite case, the VSP model can be cited from a different study or as a personal communication from its author.

Ln 218-.. The description of seismic catalog update appears too detailed and technical and overlaps with the similar section in Methodology. Please consider unifying, making it more clear and concise.

Another point concerning locations is the (mis)fit of the hypocenters with the borehole trace. IN the depth sections of Fig. 3 it appears that most hypocenters lie below the borehole trace, which is rather unlikely. Please compare e.g. Fig. 3 in Kwiatek et al (2019) where the hypocenters occur almost symmetrically

around the borehole.

Ln 364 It is interesting that the post-stimulation seismicity does not show any systematic migration. This observation should be supported by a sort of distance-time or coordinate-time plot. In fact, even the existing papers of Kwiatek and Hillers on the Helsinki stimulation do not show such data.

Ln 381 To see the events at perimeter these should be shown on top of the others, e.g. in grey

Ln 393 Please argue for the highest expected pore pressure perturbation at the bottom of the permeable zone

Ln 400 The depthward migration is not visible in Fig. 3. And further, it is very unlikely that water would flow down in the expected lithostatic conditions of the rock formation where no open fractures are expected. On the contrary, water tends to flow up due to the buyoancy effect cause by the difference in density of water and rock.

Ln 442 In the Summary, the authors mention seismic catalog as a result of the study provided to he community. This sounds great, however I would welcome to see some quality analysis of the catalog, at least to show the Gutenberg-Richter distribution distinguishing the original catalog, the newly detected and newly located events.

Ln 449 The statement "The temporal behavior of the poststimulation seismic moment release until bleed-off is still similar to the moment release observed during individual stimulation phases" sounds a bit vague.

Ln 474-476 According to the unclear description of some parts I am not sure if all of the coauthors did really contribute to the manuscript (by e.g. the manuscript correction indicated in the Author contribution section).

Figures

Fig S1 should be included as Fig. 1; this is much more informative than the present

Fig. 1 which could be moved to Supplements

Fig. S2 overlaps with Fig. 2 and using different time scale (absolute vs. relative) makes it different to compare. Why not combining Fig.S2 and Fig.2 in a single plot?

Fig. 2 is missing reference in the text. The caption does not explain the meaning of time - from which moment the days are counted? It is also not clear why you do not show also the time period during the stimulation as indicated in the manuscript title and also shown in Fig. 3.

Fig. 3 The caption should be better specified; e.g. mentioning the name OTN3 of the borehole is missing and the legend does not explain the colored bands along the borehole trace. Are these the stimulated sections and should their color correspond (at the moment it does not) to the colors of hypocenters?

Fig. 4 and 5: the yellow line is hardly visible.

Fig. 5 The three CM0 plots could be better shown with common Y axis, which would spare space and make them more legible, Also a single legend would then suffice.

Fig. 9 The black stress component are not visible enough, consider using different color.

Fig. 10 Please indicate in the caption that the stress ratio R 0.53 determined in the stress inversion is used. And shift the Px markers a bit to the right, these are very hardly visible now.

---

## Referee Comment (RC2) · Anonymous Referee #2 · 16 Nov 2020

Seismicity associated with the stimulation of an Enhanced Geothermal System in Finland is presented. A variety of analytical tools are used to extract as much information as possible. While I do not have expert knowledge of the tools, they seem to have been competently used and deliver plausible results

While I am not personally involved in EGS studies, I found the paper interesting because it relates to work that my group is doing with regard to seismicity induced by mining and the flooding of worked-out mines, as well as shale gas development.

I failed to meet the review deadline, for which I apologise. I downloaded a copy of the manuscript when I accepted the invitation, but found that I was unable to access the

supplementary material when I reviewed the paper on 14 November. I hope that the supplements are of the same high standard, but I have not checked this.

Generally the paper is well-written. There are some minor grammatical errors that I have indicated on the attached annotated version of the manuscript. The referencing style is also inconsistent - some paper titles are in Sentence case, others in Title Case.

There are also a few instances where I found the discussion difficult to follow or figures difficult to interpret. I have highlighted these and offered suggestions for improvement.

Please also note the supplement to this comment:
https://se.copernicus.org/preprints/se-2020-139/se-2020-139-RC2-supplement.pdf

─────────────────────────────

**Supplement:**

[revised manuscript text omitted]

---

## Author Comment (AC1) · 14 Dec 2020

**Response to reviewer RC1 comments: se-2020-139-RC1**

We thank the anonymous reviewer RC1 for the thoughtful review of our manuscript. The constructive comments helped us to further improve the manuscript.
We edited the manuscript carefully and addressed all comments of reviewer RC1. Please find below the detailed reply to the comments.

All reviewer comments are shown and highlighted as bold text, followed by our answers as indented normal text. Line numbers in our response refer to the tracked revised manuscript.

Please find also attached at the end of this document, the description of our data publication with further details about the seismic catalog reprocessing and its properties. This data publication document will be available with the seismic data catalog through GFZ data services: http://dataservices.gfz-potsdam.de/portal/ as a separate data publication.

General comments of Reviewer RC1

> **1) The study deals with very interesting data on injection induced seismicity in a unique experiment and gives some valuable results. These are in particular the extended catalog, the focal mechanisms and principal stresses. Providing these data to the scientific community will undoubtly help better understanding the induced seismicity in geothermal projects in hard rocks. However, despite of reasonable language (as I can assess as non-native speaker), the study is not easy to read. This holds e.g. to the parts on catalog methodology and results, which is not easy to understand. One of the reasons is structuring the paper to Methodology and Results sections. It is a good approach in general, but in some cases it breaks the individual topics and makes the paper longer and understanding more difficult. So I recommend to describe only the more sophisticated methods like 2.3, 2.3 and the location part of 2.1.**

> Authors:
> Thank you, we followed your suggestion and focused for the Methodology part only on the location paragraphs of 2.1, on section 2.3 and also on section 2.4 (we assume that this is meant under the second "2.3" in the comment). Nevertheless, we decided not to exclude the first paragraph of the Methodology part (lines 94-101) because it is a short overview of the stimulation and an introduction to the section. To not describe the seismic catalog with too much detail (as mentioned in comment #12 below) and also to avoid repetition with the Results part we deleted the second and third paragraph from the Methodology.

> A description of the seismic catalog, especially its reprocessing and its properties, has now been moved to the data publication to keep the manuscript more focused on the seismological study. Finally, we added the following short explanation in lines 102-106:

> *"The reprocessed seismic catalog with description of its properties is available as separate data publication (see section data availability) and consists of 5,456 events that were detected and located during and after the stimulation (industrial monitoring) and reprocessed in our study. A total of 55,707 smaller events were further detected during and after the stimulation but were not located or processed later on. These were also included in published seismic catalog. For further explanation about the original seismic catalog see Kwiatek et al. (2019)."*

We also excluded the entire section 2.2 from the Methodology part, but added the last two sentences of this section as an introduction to section 3.3 in the Results (lines 292-294):

*"For the spatial distribution of the seismic moment, the area around the injection well was separated into horizontal bins of 50x50 m. The cumulative seismic moment of all events within each bin was then investigated by disregarding the depth."*

To still mention the numbers of absolute and relocated stimulation and post-stimulation events included in the catalog, we modified the following sentence in lines 120-122:

*"The enhanced sub-catalog of 5,456 events including 946 post-stimulation events was reprocessed applying a new updated 1D layered velocity model developed from P-wave onset times of calibration shots obtained during a post-injection VSP campaign (Fig. S1, see also data publication)."*

Lastly, we also updated the numbers of events included in the catalog in lines 132-138:

*"A total of 2,958 reprocessed events were absolute located around the injection well OTN-3 at an epicentral distance of less than 5 km and at depth of 4.5 to 7 km. The hypocenters of these events were included to the reprocessed and published catalog.*

*        To further refine the quality of hypocenter locations, 2,178 from the 2,958 absolute located events with at least 10 P-wave and 4 S-wave picks were selected and the double-difference relocation technique (hypoDD) was applied using the new VSP-derived velocity model (Waldhauser and Ellsworth, 2000)."*

We hope that these changes help to improve the understanding and simplify the reading of the paper.

**2) I also think that the spectrum of methods applied is too wide with no clear focus. The authors should decide if they present new high quality extensive seismic catalog whose parameters are characterized by a set of suitable (statistical) methods or they present a seismological study including interpretations. The point is that despite the catalog is the most valuable output, it is never characterized by at least Gutenberrg-Richter distribution and similar methods.**

Authors:
We decided to keep the description of the new catalog to minimum and shift discussion on its preparation to the separate data publication (please see the data publication document attached at the end of our responses). In consequence, the methodology and processing parts of the manuscript were streamlined, and we focused our analysis on the source mechanisms and mechanisms complexities, so we now believe the focus of the manuscript was sharpened.

**3) The authors also spent a lot of effort determining focal mechanisms using quite sophisticated method to get maximum number of mechanisms, they however do not show the whole set of FM and assess their quality.**

Authors:
The quality of focal mechanisms was assessed by the root mean square fault plane uncertainties of the estimated focal mechanisms (Hardebeck and Shearer, 2002). We only further investigated focal mechanisms which had uncertainties less or equal 35°, as suggested by Hardebeck and Shearer (2002). Focal mechanisms with associated uncertainties are a part of the data publication, and we indicated this in the text in lines 196-198:

*"The final catalog of focal mechanisms includes 191 events with either manually or estimated polarity pattern and is presented with associated uncertainties in the data publication (see section data availability)."*

**4) I am also not sure about the improved quality of locations in terms of their asymmetric position to the borehole.**

Authors:
The asymmetric distribution of hypocenter locations to the borehole is indeed interesting, but we are at the moment very confident that this is the case. This is supported by two independent analyses, one conducted by the main Author and one being a part of new study by Kwiatek et al. (2021).

We identified that the position of the cluster is affected predominantly by the assumed $V_P/V_S$ ratio. Thus, as the positioning of the cluster was vital for the interpretation of the seismicity, we optimized the cluster position using two criteria: 1) the sum of residuals for all events from location procedure should be minimal, and 2) hypocenters of events at the beginning of stimulation should occur in direct vicinity of injection interval. Our analysis, as presented in submitted manuscript, resulted in $V_P/V_S$ ratio optimized to 1.67 (which is not very different from 1.68 assumed in Kwiatek et al., 2019). However, we updated $V_P/V_S$ ratio to 1.71 using new seismic catalog obtained during 2020 stimulation in *OTN-2* well (which is a subject of a pending study of Kwiatek et al., 2021). The new defined constrain was that 3) events from 2020 stimulation should cluster around *OTN-2* well. The final outcome of locations is shown in the Figure below (however, we restrain from presenting 2020 stimulation data in SE manuscript, as this is a part of pending study).

[Figure]

The revised manuscript uses now hypocenters estimated with a $V_P/V_S$ of 1.71. We updated our seismic data catalog.

Using the higher ratio of 1.71, the hypocenters of the 2018 induced events are shifted approximately 300 m upwards in depth. With this shift in depth, the hypocenters are also now more symmetrically located around the injection well, as shown in the revised manuscript (updated Fig. 3a-b). Because of minor changes in takeoff angles, no significant change in focal mechanisms was observed.

We also updated the following sentence in lines 123-126:

"*Thus, the $V_P/V_S$ ratio was optimized by a trial-and-error procedure, where we ultimately constrained a $V_P/V_S$ ratio of 1.71 that minimized the cumulative residual errors of all located events, and at the same time kept the first induced events close to corresponding injection well OTN-3.*"

**5) As a result I believe the paper should be restructured according to its main focus - presentation of new data. Details of my comments which should be adressed in a major revision are summarized below.**

Authors:
We restructured the manuscript, especially the Methodology and Results parts (for the results, please see the response to comment #12), to focus on the seismological study while keeping the development and properties of the catalog to the minimum. Associated data publication (please see the attached document at the end) contains relevant information on how the catalog was designed and catalog properties.

Particular comments of Reviewer RC1

**6) Ln 109-120 (Methodology). The explanation about different subsets of larger and smaller events and their relocation is not very clear. E.g. how many events were above Mw 0.7; were the 3464 events chosen from this subset?; did these events occur during stimulation because you added 321 post-stim events?; did 68 events com from this subset?...**

Authors:
With updating our seismic catalog using now a $V_P/V_S = 1.71$, we also simplified the selection of events used for reprocessing, especially not distinguishing between subsets of larger and smaller events anymore. The reprocessing steps and details about the seismic catalog and its statistical properties are now part of the data publication.

**7) Ln 172 - 176. Please explain the SVD application in more detail. The point is that SVD is usually used to find a common pattern in a data set. For this you would need more polarity patterns for each event that just one, which you have as a result of cross correlation. The next question is whether the polarity matrix (eq. 2) shows the polarity fit between the target and template events as indicated on Ln 171 or the fit of polarities themselves. In the first case, it could not be used for calculating focal mechanisms.**

Authors:
Indeed, the SVD is usually used to find a common pattern in a data set and this is also the reason why we applied the SVD. The method of Shelly et al. (2016) is a well-established approach were the SVD is applied to extract a common polarity signal from

a matrix that contains the obtained relative polarities between each target events and all template events, considering each station and phase (in our case only P-phase) separately.

For each station, the left singular vector is obtained by applying the SVD to the above mentioned matrix. This vector provides a means of estimating the most consistent set of polarities (sign of the elements) for each target event and station (Shelly et al., 2016).

In our manuscript, the left singular vectors of all stations are presented in the columns of the matrix in equation 2. Therefore, only the most reasonable polarity for each target event and each station is presented in equation 2 as a best fit of many relative polarities derived from cross-correlation between this target event and many templates. Thus, the best fit for each target event still shows a polarity ambiguity. This sign ambiguity of polarities can only be resolved later on when considering the manually picked polarities of some target events.

We restrain from describing the methodology in manuscript in details, as this is a subject of Shelly et al. (2016) where the method is described in details in step-by-step fashion.

We added the following sentence to the manuscript in lines 180-181:

*"For each station k, the vectors containing relative polarity estimates between one target event i and all templates j were gathered in a i-by-j matrix."*

We further rewrote the following sentence in lines 182-185:

*"A Singular Value Decomposition (SVD) was applied to the relative estimated polarity matrix of each station k to extract the strongest common signal of any target event obtained by the first left singular vector of the SVD (Shelly et al., 2016; Rubinstein and Ellsworth, 2010)."*

**8) Ln 178. The way you reduced the polarity ambiguity is not clear; by considering manually picked events one can verify the automatic picks, I believe.**

Authors:
This is precisely what we have performed. Manually picked events and their "true" polarities were used to resolve the ambiguity of SVD-derived polarities for all events at each station, separately. If the SVD-derived polarities has the same sign as the manually picked polarities for one station, than all automatically derived polarities of the other events should also have the right polarities for this particular station due to the first singular vector of the SVD.

We updated the following part of the manuscript (lines 190-192):

*"For each station, the SVD-derived polarities of these events were compared with manually picked polarities to investigate whether the polarities have similar or opposite signs. In case of same polarities, the SVD-derived polarities of other events should also show the right sign for the particular stations."*

**9) Ln 185. The final sentence mentioning the resulting reverse faulting fits rather to the Results than Methodology section.**

Authors:
Yes, we agree. We deleted this sentence at the end of our Methodology section.

**10) Ln 195. Please argue for using this distance metrics - what is the reason for 1.5 in the denominator? And which type of cluster analysis did you use? What is the difference to the published method of moment tensor clustering of Cesca (2014)?**

Authors:
The choice of 1.5 is only to scale the value to range 0-1 (as the Kagan rotation angle $\theta$ ranges 0°-120°, our distance metrics $PR_{ij}$ scale from 0 to 1). The cosine was used to rescale Kagan rotation angles and to emphasize large differences in $\theta$. We found for our dataset that this choice does not influence the discussed clustering outcome (i.e. one could use distance metrics based on the Kagan angle $\theta$ alone).

As stated in the manuscript, we used well-established hierarchical cluster analysis with distance measured using average distance (Unweighted average distance, UPGMA) and Euclidean distance metrics. The selection of particular distance metrics between clusters was made objectively using the one with highest value of the cophenetic correlation coefficient. Cesca et al. (2014) applied a density-based clustering technique *DBSCAN* (Ester et. al, 1996). Clusters can be identified as densely populated "areas" with a much higher number of points than outside of a presumable cluster. Cesca's approach is more general, as it can be used for non-DC sources. However, in case of pure DC moment tensors, a distance metric based on the Kagan angle alone is used by Cesca et al. (2014), which is comparable to our case.

**11) Ln 209-216 (Results). I think that the VSP based model deserves more attention. The present way is not appropriate - to show the model as a result without any more details. If it is considered as a result of this study, the data, methods and results should be shown. In the opposite case, the VSP model can be cited from a different study or as a personal communication from its author.**

Authors:
Following Reviewer suggestion, we separated detailed description on catalog development from mechanism complexity analysis. We added details about the VSP velocity model build-up to the data publication. In manuscript we switched Fig. 1 with Fig. S1, as suggested in comment #20. We also added the following sentences to the caption of the new Fig. S1:

*"The VSP-derived velocity model shows a velocity inversion between 3 and 6 km depth. Below this velocity inversion, a constant velocity of 6 km s$^{-1}$ is suggested from sonic logs which were used for velocity estimation between 5.1 km and 6.4 km depth."*

**12) Ln 218-... The description of seismic catalog update appears too detailed and technical and overlaps with the similar section in Methodology. Please consider unifying, making it more clear and concise. Another point concerning locations is the (mis)fit of the hypocenters with the borehole trace. In the depth sections of Fig. 3 it appears that most hypocenters lie below the borehole trace, which is rather unlikely. Please compare e.g. Fig. 3 in Kwiatek et al (2019) where the hypocenters occur almost symmetrically around the borehole.**

Authors:
We shortened and restructured the entire section "Seismic catalog update" in the Results part to make it more unified with the Methodology part. A description of the seismic catalog and its reprocessing has now been moved to the data publication. We kept the discussion related to post-stimulation events, as these were not analyzed yet by Kwiatek et al. (2019) or by Hillers et al. (2020).

As mentioned in the response of comment #4 above, by using the updated catalog with a $V_P/V_S$ ratio of 1.71, the hypocenters are now more symmetrically located around the borehole trace (Fig. 3b) and no longer below as it was the case using a $V_P/V_S$ ratio of 1.67.

**13) Ln 364. It is interesting that the post-stimulation seismicity does not show any systematic migration. This observation should be supported by a sort of distance-time or coordinate-time plot. In fact, even the existing papers of Kwiatek and Hillers on the Helsinki stimulation do not show such data.**

Authors:
Thank you for this comment. We have produced a distance-time plot for the entire stimulation including all separate phases. For each event we took the shortest distance to the open-hole section of the injection well. For phases 1 and 2 we find relative fast migration to roughly 200 m to the well. Starting with phase 3 some events indicate migration out to 400 m distance to the well, but not further. This holds for the post-stimulation phase. The diagram is added to the data publication.

[Figure]

**14) Ln 381. To see the events at perimeter these should be shown on top of the others, e.g. in grey.**

Authors:
We updated Fig. S4 (in the revised manuscript S3) by plotting the events with $M_W \geq 1$ which occurred during the stimulation in dark grey on top of all relocated events (light

grey) to highlight the narrow zone. We further color-coded events with $M_W \geq 1$ which occurred after the end of stimulation in orange to indicate that these events are located at the perimeters of the narrow zone.

We also added to the caption of this Figure the following sentence:

"Events with $M_W \geq 1$ that occurred during and after the stimulation are color-coded as dark grey and orange, respectively."

**15) Ln 393. Please argue for the highest expected pore pressure perturbation at the bottom of the permeable zone.**

Authors:
(see also reply to comment #16). The largest pore pressure perturbation is simply expected to be at or close to the well and will progressively decrease with increasing distance. Updated seismic catalog shifted events to shallower depths so they are not significantly deeper than the bottom-hole of injection well *OTN-3*. Thus, the highest seismicity activity and largest seismic events are not anymore at the "bottom of the permeable zone", but are correlated to the bottom-hole of the injection well *OTN-3*. It is expected that this area is characterized by highest pore pressure perturbation, as this is where injection was performed in stages 1-3. Attached here is the figure from data publication showing relation between magnitude and depth.

We replaced "bottom" with "deepest" zone in the referred sentence, pointing out to the fact that largest events occur in the bottom cluster.

[Figure]

**16) Ln 400. The depthward migration is not visible in Fig. 3. And further, it is very unlikely that water would flow down in the expected lithostatic conditions of the rock formation where no open fractures are expected. On the contrary, water tends to flow up due to the buyoancy effect cause by the difference in density of water and rock.**

Authors:
We agree with the reviewer that such behavior is quite unexpected, although it is observed in some highly fractured reservoirs (see e.g. Kwiatek et al., 2015, Kwiatek et al., 2018). However, the updated seismic catalog with new $V_P/V_S$ ratio effectively

shifted all events to the shallower depths, rendering original comment on depth migration doubtful. It is still visible that in later stages the seismicity in the bottom cluster tends to locate at larger depths (see previous figure), but the depth of later events is not significantly exceeding the depth of bottom hole of *OTN-3*. This restrained us from suggesting that water flows down, and we suggest that occurrence of seismicity is simply related to pore pressure perturbation that is stronger around the bottom part of injection well *OTN-3*.

**17) Ln 442. In the Summary, the authors mention seismic catalog as a result of the study provided to the community. This sounds great, however I would welcome to see some quality analysis of the catalog, at least to show the Gutenberg-Richter distribution distinguishing the original catalog, the newly detected and newly located events.**

> Authors:
> We include the description of the seismic catalog and their properties to the data publication. Besides of providing details about the catalog reprocessing in the data publication, we include there statistical and spatio-temporal properties of developed catalog.

**18) Ln 449. The statement "The temporal behavior of the post-stimulation seismic moment release until bleed-off is still similar to the moment release observed during individual stimulation phases" sounds a bit vague.**

> Authors:
> We rewrote the sentence in the Summary and conclusions part:
>
> *"Until shortly after the bleed-off, the increase in the cumulative moment release of the post-stimulation seismicity with time is comparable with the slope of the $CM_0$ during individual stimulation phases but substantially less afterwards. This is especially observed for the seismicity of the deepest hypocenter cluster."*

**19) Ln 474-476. According to the unclear description of some parts I am not sure if all of the coauthors did really contribute to the manuscript (by e.g. the manuscript correction indicated in the Author contribution section).**

> Authors:
> We state the Author contribution as follows:
>
> *"M.L.: data reduction, analysis and results interpretation, draft version of the manuscript, and associated data publication. G.K. and P.M.-G.: data analysis, results interpretation, and manuscript correction. M.B., G.D., and P.H.: results interpretation and manuscript correction. T.S.: project management, drilling and stimulation program development and managing, and manuscript correction."*

Comments of reviewer RC1 to the Figures

**20) Fig S1. should be included as Fig. 1; this is much more informative than the present Fig. 1 which could be moved to Supplements.**

> Authors:
> We swapped Fig. S1 and Fig. 1.

**21) Fig. S2 overlaps with Fig. 2 and using different time scale (absolute vs. relative) makes it different to compare. Why not combining Fig.S2 and Fig.2 in a single plot?**

Authors:
Thank you for this suggestion, we combined both Figures to a new Fig. 2 using an absolute time scale. We therefore updated the following sentences in lines 237-239 in the manuscript:

"*The moment magnitudes of the absolute located and relocated seismicity is plotted with time during and after shut-in as grey and orange dots in Fig. 2. The five different stimulation phases (P1-P5) performed in 2018 are also shown in Fig. 2 in combination with the wellhead pressure and seismic event rate.*"

**22) Fig. 2 is missing reference in the text. The caption does not explain the meaning of time - from which moment the days are counted? It is also not clear why you do not show also the time period during the stimulation as indicated in the manuscript title and also shown in Fig. 3.**

Authors:
Thank you for mentioning the missing reference of Fig. 2. With combining Fig. 2 and Fig. S2 to a new Fig. 2 in the revised manuscript, the reference for Fig. 2 is now mentioned in line 238. For the updated Fig. 2, absolute times (in days) are now used for a better understanding.

Initially we wanted to keep the focus on the post-stimulation seismicity in the original Fig. 2 because this is mainly the new data and not analyzed by Kwiatek et al. (2019) or Hillers et al. (2020) and therefore, the time period during stimulation was not shown. However, the suggestion of combining Fig. 2 and Fig. S2 is a good idea and thus the seismicity and time period during the stimulation is now also presented.
For the updated Fig. 2, we rewrote the caption as followed:

"*Stimulation protocol with moment magnitudes of induced seismicity during stimulation phases P1-P5 and post-stimulation time period. The magnitudes of absolute located and relocated events are shown as grey and orange dots, respectively. The green solid line presents the wellhead pressure during the stimulation. The seismic event rate per day is shown by the solid blue line.*"

**23) Fig. 3: The caption should be better specified; e.g. mentioning the name OTN3 of the borehole is missing and the legend does not explain the colored bands along the borehole trace. Are these the stimulated sections and should their color correspond (at the moment it does not) to the colors of hypocenters?**

Authors:
We specified the caption by adding the name of the injection well *OTN-3* and explaining the color bands along the borehole trace of *OTN-3*.

We apologize for the confusion about the colored bands along the borehole trace. Unfortunately, the colors along *OTN-3* were wrongly plotted in Fig. 3. We updated the colors which are now corresponding to the colors of the five stimulation stages.

For a better visibility, we also changed the color of the stimulation phase P5 hypocenters to a darker yellow.

**24) Fig. 4 and 5: the yellow line is hardly visible.**

Authors:
We changed the color to a darker yellow in both Figures.

**25) Fig. 5: The three CM0 plots could be better shown with common Y axis, which would spare space and make them more legible, also a single legend would then suffice.**

Authors:
Thank you for this suggestion. We updated the Figure using one common y-axis and one legend for all three subplots now.

**26) Fig. 9: The black stress component are not visible enough, consider using different color.**

Authors:
We now use white as color for the stress component marker symbols and the marker text.

We updated the sentence in the caption of Fig. 9:

*"White upward and downward pointing triangle represent maximum and minimum principal stress axes $\sigma_1$ and $\sigma_3$, respectively."*

**27) Fig. 10: Please indicate in the caption that the stress ratio R 0.53 determined in the stress inversion is used. And shift the Px markers a bit to the right, these are very hardly visible now.**

Authors:
We added the following sentence to the caption of Fig. 10:

*"A stress ratio of R = 0.53 was used for stress inversion."*

For a better visibility, we also shifted the text of the P1 and P2 markers a bit further outside of each marker symbol.

References not used in the manuscript

Cesca, S., Ali, S., and Dahm, T.: Seismicity monitoring by cluster analysis of moment tensors, Geophysical Journal International, 196, https://doi.org/10.1093/gji/ggt492, 2014.

Kwiatek, G., Martínez-Garzón, P., Plenkers, K., Leonhardt, M., Zang, A., Specht, S., Dresen, G., and Bohnhoff, M.: Insights into complex subdecimeter fracturing processes occurring during a water injection experiment at depth in Äspö Hard Rock Laboratory, Sweden, Journal of Geophysical Research: Solid Earth, https://doi.org/10.1029/2017JB014715, 2018.

**Data publication related to**

**"Seismicity during and after stimulation of a 6.1 km deep Enhanced Geothermal System in Helsinki, Finland"**

Maria Leonhardt, Grzegorz Kwiatek, Patricia Martínez-Garzón, Pekka Heikkinen

**1. Structure of seismic catalog file**

Column 1:

ID number of event

If events were detected but not located, the ID is 0.

Column 2:

Datenumber (integer part = day since year 0)

Column 3-8:

Year, month, day, hour, minute, second

Column 9:

Local "Helsinki" magnitude $M_{LHEL}$

Column 10:

Moment magnitude $M_W$

Column 11-13:

Easting (m), northing (m), altitude (m) of absolute location

Column 14-16:

Easting (m), northing (m), altitude (m) of relocation

Column 17-19:

Strike, dip, rake of preferred nodal plane from estimated focal mechanisms

Column 20:

Root mean square fault plane uncertainties of estimated focal mechanisms

For further details about the reprocessing of the catalog and its properties, please see section 2 and 3 below.

**2. Seismic catalog development**

The original seismic catalog created during stimulation campaign has been reprocessed by Kwiatek et al. (2019), and included 6,150 located and ~54,000 detected earthquakes.

In first step, this catalog was extended in time to cover the post-stimulation period of 63 days. In the following, we selected best quality 5,456 events that were located during and after the stimulation, and reprocessed them in our study, as discussed in details below. The original catalog of detections was reviewed as well, resulting in 55,707 smaller events detected during and after the stimulation. Thus the total seismic catalog presented in this data publication contains 61,163 earthquakes in the period of 112 days that occurred in the vicinity of the *OTN-3* well. In the following sections we present the development of seismic catalog.

**2.1 Seismic network**

Following Kwiatek et al., (2019), the real-time telemetered network monitoring the stimulation campaign was composed of 24 borehole seismographs, fabricated, installed, and operated by Advanced Seismic Instrumentation and Research (www.asirseismic.com). The 12-level borehole array of three-component 15-Hz natural frequency Geospace OMNI-2400 geophones was sampled at 2 kHz and placed at depths of 1.95 to 2.37 km in the *OTN-2* well. Additional 12-station three-component $f_N$ = 4.5 Hz Sunfull PSH geophones sampled at 500 Hz were installed in 0.30- to 1.15-km-deep wells. These surrounded the project site at 0.6- to 8.2-km epicentral distances. These two networks were operating months before the start of stimulation with no event detected in the vicinity of *OTN-3* injection well. Data from these 24 sensors were used in processing of seismic data forming the data publication.

**2.2 Detection catalog**

We followed the same approach as presented in Kwiatek et al. (2019). P-wave arrivals unused in locations, but detected using the array located in *OTN-2* well, were further analyzed. Assuming that a small event that is detected solely at the *OTN-2* array must occur in its immediate vicinity, we placed a hypothetical seismic source at the bottom of *OTN-3* where the injection took place. We then calculated travel times of P-waves to the sensors forming the *OTN-2* array, obtaining a particular pattern (offset) of expected P-wave arrivals at these stations. We then scanned the catalog of unused *OTN-2* P-wave arrivals for this particular pattern, and each matching set of detections was attributed to an event occurring in the vicinity of the *OTN-3* well. The magnitude (see section 2.5) was calculated assuming that the event occurred at the bottom of the *OTN-3* injection well. This procedure allowed us to enhance the catalog by 55,707 earthquakes.

**2.3 VSP-based velocity model**

In original study of Kwiatek et al. (2019), the 1D velocity model based on velocity logs was used. In the study of Leonhardt et al. (2020), the new velocity model was developed from P-wave onset times of calibration shots obtained during a post-injection Vertical Seismic Profiling (VSP) campaign.

The VSP campaign was performed in October 2018 after the end of the stimulation. Overall, 47 calibration shots were performed at 7 shot points located around the injection well *OTN-3* with a maximum distance of less than 8 km. Shot points were prepared with explosives in holes up to 40 m depth. The VSP campaign was monitored by a

17-level vertical chain with 3-components geophones located in the injection well *OTN-3* in a depth between 2.5 km and 4.5 km. In addition, the 12-level vertical geophone chain, used for the stimulation, was also monitoring the VSP shots to cover the depth above 2.5 km.

The 1D velocity model used in Leonhardt et al. (2020) was developed from the data of VSP shot performed close to the OTRA station. This secured that wave propagation ray was nearly vertical between the shot location and seismic arrays. This allowed us to convert travel-path velocities calculated at different sensors forming the array along the *OTN-3* well to interval velocities of the 1D velocity model. For the depths below 4.5 km which was not covered by seismic rays of VSP shots we used information from sonic logs, available depth between 5.1 km and 6.4 km. The velocity model is presented in Fig. 1. Due to a low Signal-to-Noise (S/N) ratio of the VSP data, the S-wave arrival times could not be determined.

The 1D VSP-derived velocity model shows a velocity inversion between 3 and 6 km depth (Fig. 1). The maximum P-wave velocity is 0.15 km s$^{-1}$ larger than the maximum velocity modelled by Kwiatek et al. (2019) where a constant velocity of 6.4 km s$^{-1}$ starting at 3 km depth was assumed. Below the velocity inversion, approximately constant velocity of 6 km s$^{-1}$ is suggested from sonic logs for the updated 1D velocity model (Fig. 1).

[Figure]

**Figure 1**. Comparison of 1D velocity model developed from VSP profiling (solid line) and the one used in Kwiatek et al. (2019).

**2.4 Earthquake location and relocation**

The sub-catalog of 5,456 events was reprocessed applying the new 1D layered velocity model. Thus, the $V_P/V_S$ ratio had to be optimized by a trial-and-error procedure, as discussed in Leonhardt (2020). We found the optimum $V_P/V_S$ by minimizing the cumulative residual errors of all located events while keeping first induced seismic events close to the

injection well *OTN-3*. The optimized $V_P/V_S$ ratio of 1.71 was therefore selected which is similar to that used in Hillers et al. (2020).

The hypocenter locations were estimated using the Equal Differential Time (EDT) method (Zhou, 1994; Font et al., 2004; Lomax, 2005) and the new VSP-derived velocity model. In addition, station corrections were applied. The minimization of travel time residuals:

$$\left\lVert \left(T_j^{th} - T_i^{th}\right) - \left(T_j^{obs} - T_i^{obs}\right) \right\rVert_{L_2} = min, \tag{1}$$

where $T^{th}$ and $T^{obs}$ are all unique pairs (i,j) of theoretical and observed travel times of P- and S-phases, were resolved using the Simplex algorithm (Nelder and Mead, 1965; Lagarias et al., 1998) . A total of 2,958 reprocessed events were absolute located around the injection well *OTN-3* at an epicentral distance of less than 5 km and at depth of 4.5 to 7 km. The hypocenters of these events were included to the reprocessed and published catalog.

To further refine the quality of hypocenter locations, 2,178 from the 2,958 absolute located events with at least 10 P-wave and 4 S-wave picks were selected and the double-difference relocation technique (hypoDD) was applied using the new VSP-derived velocity model (Waldhauser and Ellsworth, 2000). An iterative least-square inversion was used to minimize residuals of observed and predicted travel time differences for event pairs calculated from the existing P- and S-wave picks of the selected catalog data. The residuals were minimized in ten iterations steps. For the last iteration, the maximum threshold for travel time residuals were set to 0.08 s and the maximum distance between the catalog linked event pairs was defined as 170 m. With the hypoDD method 1,986 events were relocated and thus 91 % of the selected 2,178 events. The residuals of the relocations have a root mean square error of 9 ms. The relocation uncertainties were then assessed using a bootstrap technique (Waldhauser and Ellsworth, 2000; Efron, 1982) leading to relative location precision not exceeding ±52 m for 95 % of the catalog.

**2.5 Basic source characteristics and statistical properties**

Local "Helsinki" magnitude $M_{LHEL}$ has been calculated from ground displacement seismograms integrated from ground velocity records (Uski and Tuppurainen, 1996; further updated by Uski et al. (2015) to smaller events). The magnitude was calculated separately on each station (24 sensors) using vertical component seismograms, and then averaged. The moment magnitudes of all events were estimated from local magnitudes $M_{LHEL}$ using formula from Uski et al. (2015). The seismic moment was recalculated from $M_W$ using formula of Hanks and Kanamori (1979).

The magnitude of completeness $M_C$ as well as the *b*-value were calculated assuming a Gutenberg-Richter (GR) power law: $\log_{10} N = a - bM_C$, where $N$ is the cumulative number of earthquakes with magnitudes larger than $M_C$. Following the Goodness-of-fit method (Wiemer and Wyss, 2000), the magnitude of completeness and the *b*-value were estimated assuming that the GR power law can fit 98 % of the seismic data.

**3. Seismic catalog properties**

The reprocessed seismic catalog covers the time period between 4th of June and 24th of September 2018. The stimulation was performed during the first 49 days. After shut-in of injection, i.e. after 22nd of July 2018 at 15:52 UTC, further 63 days of the post-stimulation time period were monitored.

Overall, 61,163 earthquakes were detected during and after the stimulation. From the 55,707 events that were detected but not further processed, 52,107 detections occurred during the stimulation whereas another 3,600 detections were monitored after the stimulation. From the 5,456 events that were further processed, 4,510 events were monitored during and 946 events were monitored after the end of stimulation.

**3.1 Moment magnitudes**

The 55,707 event detections, that were not further located or processed, had moment magnitudes between $M_W = -0.95$ and $M_W = 1.53$. The subset of 2,958 events that were absolute located within the target volume around the injection well *OTN-3* with an epicentral distance of less than 5 km and a depth between 4.5 km and 7 km had moment magnitudes between $M_W = -0.84$ and $M_W = 1.87$. The 213 post-injection events that were absolute located within the target volume around *OTN-3* showed a minimum moment magnitude of $M_W = -0.69$. The largest observed magnitude was $M_W = 1.54$ for the absolute located post-stimulation events. The subset of 1,987 relocated events showed moment magnitudes between $M_W = -0.49$ and $M_W = 1.87$. The 70 relocated post-stimulation events had a minimum magnitude of $M_W = -0.07$.

**3.2 Relocated catalog**

Figure 2 presents the relocated seismicity which occurred in three spatially separated clusters elongated in southeast (SE) - northwest (NW) direction and centered along the injection well, in good agreement with Kwiatek et al. (2019). Elongation of the clusters in SE-NW direction is sub-parallel to the local maximum horizontal stress $S_H^{max} = 110°$ (Kwiatek et al., 2019). Further details about the relocated seismicity are discussed in Leonhardt et al. (2020).

[Figure]

[Figure]

**Figure 2.** Hypocenters of relocated events. (a) Map view and (b) SW-NE depth section. The hypocenters are color-coded with the stimulation phases (cf. Kwiatek et al., 2019) and size corresponds to moment magnitude. Relocated seismicity that occurred after the stimulation is represented as grey dots. The five injection stages are marked as color bands along the borehole trace from the bottom of the open-hole toward the casing shoe of the injection well *OTN-3* (black). The new *OTN-2* well (grey) was drilled in 2019 to 2020 after the stimulation.

**3.3 Spatio-temporal characteristics**

Figure 3 shows the development of the seismicity with the horizontal distance from injection well *OTN-3*. This shows quick expansion of seismicity in lateral direction (mostly along SE-NW direction) in first two stimulation phases P1-P2 lasting 20 days (cf. Kwiatek et al., 2019), where the injection rates and injection well head pressures were the highest (cf. Kwiatek et al., 2019; Leonhardt et al., 2020). In following stimulation phases P3-P5, the expansion is slower and the seismicity front reaches approx. 400 m horizontal distance from *OTN-3* well. The post-stimulation phase displays no signatures in propagation with scattered seismicity confined to 400 m horizontal distance from *OTN-3* well.

[Figure]

**Figure 3**. Spatial development of the seismicity with time during *OTN-3* stimulation (until day 49) and in post-stimulation phase (from day 49). For each event, the distance is calculated as a distance between earthquake epicenter (EASTING, NORTHING) and the coordinate of the *OTN-3* well (EASTING, NORTHING) at the depth of earthquake (horizontal distance). The red, blue and green curves represent expected space-time evolution of a fluid pressure perturbation front triggering seismicity assuming that it is solely controlled be scalar fluid pressure diffusion in a homogeneous isotropic medium (e.g. Shapiro et. al., 2020).

Figure 4 presents the dependence between earthquake depth and local magnitude. The figure marks the three distinct clusters of seismicity (cf. Fig. 2) developed during hydraulic stimulation. Largest seismic events as well as the highest level of seismic activity is observed in the lowermost cluster. This is expected due to expected elevated pore fluid pressures in the direct vicinity of injection activities, suggesting the seismic activity, as well as maximum magnitude is pressure-controlled (cf. discussion in Kwiatek et al., 2019; Bentz et al., 2020; Wang et al., 2020a, 2020b).

[Figure]

**Figure 4**. Dependence between earthquake depth (here presented as altitude a.s.l.) and local magnitude.

**3.4 Gutenberg-Richter distribution**

The catalog combining locations and detections displays $b = 1.25$ with magnitude of completeness $M_C = -1.10$ (Figure 5). Above $M_{LHEL}$ 1.5 the statistically significant roll-off is visible which was attributed to either geometrical constraint on pre-existing fracture network or limitation to fault strength (cf. Kwiatek at al., 2019). We note that although the magnitude of completeness of the full catalog is $M_C = -1.10$, the day-night cycles and associated anthropogenic noises reduces the completeness by approx. 0.2 (cf. Figure 2 in Kwiatek et al. (2019) where day-night cycle is clearly visible).

However, processing of events with $M_{LHEL} < -0.7$ should be performed with caution. In a pending study (G. Kwiatek – pers. comm.) we note local magnitude estimates of small events with $M_{LHEL} < -0.7$ are affected by high-frequency noises above 60 Hz (multiple resonance peaks) observed on sensors forming the vertical array in *OTN-2* well. The origin of these noises has been correlated to technological activities at the injection site, with the most likely noise source attributed to the high-performance injection pumps, as the noise seem to be correlation to injection rates. As recordings from *OTN-2* arrays are used to calculate local magnitude of smaller events that are not detected using the sensors close to the surface, and the local magnitude is calculated from integrated ground displacement seismograms which further emphasize the (temporary varying and resonant) noises, we expect significant bias in estimates of $M_{LHEL}$ for $M_{LHEL} < -0.7$. This may lead to potential problems while analyzing statistical properties of induced seismicity such as magnitude correlations and or inter-event time statistics, to name a few. We suggest $M_C = -0.7$ as a safe magnitude threshold that is not affected by noises originating from technological activity and day-night cycles. The subject is a topic of pending study (G. Kwiatek – pers. comm.) and this document will be updated accordingly when new information becomes available.

[Figure]

**Figure 5**. Magnitude-frequency relation for the entire seismic catalog analyzed in Leonhardt et al. (2020).

**References**

Bentz, S., Kwiatek, G., Martínez-Garzón, P., Bohnhoff, M., and Dresen, G.: Seismic moment evolution during hydraulic stimulations, Geophysical Research Letters, 47, e2019GL086185, https://doi.org/10.1029/2019GL086185, 2019.

Efron, B.: The jackknife, the bootstrap, and other resampling plans, Siam, 1982.

Font, Y., Kao, H., Lallemand, S., Liu, C.-S., and Chiao, L.-Y.: Hypocentre determination offshore of eastern Taiwan using the maximum intersection method, Geophysical Journal International, 158, 655–675, https://doi.org/10.1111/j.1365-246X.2004.02317.x, 2004.

Hanks, T.C. and Kanamori, H.: A moment magnitude scale, Journal of Geophysical Research: Solid Earth, 84, 2348–2350, https://doi.org/10.1029/JB084iB05p02348, 1979.

Hillers, G., Vuorinen, T., Uski, M., Kortström, J., Mäntyniemi, P., Tiira, T., Malin, P., and Saarno, T.: The 2018 geothermal reservoir stimulation in Espoo/Helsinki, Southern Finland: Seismic network anatomy and data features, Seismological Research Letters, 91 (2A), 770–786, https://doi.org/10.1785/0220190253, 2020.

Kwiatek, G., Saarno, T., Ader, T., Bluemle, F., Bohnhoff, M., Chendorain, M., Dresen, G., Heikkinen, P., Kukkonen, I., Leary, P., Leonhardt, M., Malin, P., Martínez-Garzón, P., Passmore, K., Passmore, P., Valenzuela, S., and Wollin, C.: Controlling fluid-induced seismicity during a 6.1-km-deep geothermal stimulation in Finland, Science Advances, 5, eaav7224, https://doi.org/10.1126/sciadv.aav7224, 2019.

Lagarias, J.C., Reeds, J.A., Wright, M.H., and Wright, P.E.: Convergence properties of the Nelder--Mead simplex method in low dimensions, SIAM J. Optim., 9, 112–147, https://doi.org/10.1137/S1052623496303470, 1998.

Leonhardt, M., Kwiatek, G., Martínez-Garzón, P., Bohnhoff, M., Saarno, T., Heikkinen, P., and Dresen, G.: Seismicity during and after stimulation of a 6.1 km deep enhanced geothermal system in Helsinki, Finland, Solid Earth Discuss., https://doi.org/10.5194/se-2020-139, in review, 2020.

Lomax, A.: A reanalysis of the hypocentral location and related observations for the great 1906 California earthquake, Bulletin of the Seismological Society of America, 95, 861–877, https://doi.org/10.1785/0120040141, 2005.

Nelder, J.A. and Mead, R.: A simplex method for function minimization, The Computer Journal, 7, 308–313, https://doi.org/10.1093/comjnl/7.4.308, 1965.

Shapiro, S. A., Rothert, E., Rath, V., and Rindschwentner, J.: Characterization of fluid transport properties of reservoirs using induced microseismicity, Geophysics, 67(1), 212–220, https://doi.org/10.1190/1.1451597, 2020.

Uski, M. and Tuppurainen, A.: A new local magnitude scale for the Finnish seismic network, Tectonophysics, 261, 23–37, https://doi.org/10.1016/0040-1951(96)00054-6, 1996.

Uski, M., Lund, B., and Oinonen, K.: Scaling relations for homogeneous moment based magnitude, in: Evaluating seismic hazard for the Hanhikivi nuclear power plant site. Seismological characteristics of the seismic source areas, attenuation of seismic signal, and probabilistic analysis of seismic hazard, edited by: Saari, J., Lund, B., Malm, M., Mäntyniemi, P., Oinonen, K., Tiira, T., Uski, M., and Vuorinen, T., (Report NE-4459, ÅF-Consult Ltd.), 125 pp., 2015.

Waldhauser, F. and Ellsworth, W.L.: A double-difference earthquake location algorithm: Method and application to the Northern Hayward Fault, California, Bulletin of the Seismological Society of America, 90, 1353–1368, https://doi.org/10.1785/0120000006, 2000.

Wang, L., Kwiatek, G., Rybacki, E., Bohnhoff, M., and Dresen, G.: Injection-induced seismic moment release and lLaboratory fault slip: Implications for fluid-induced seismicity, Geophysical Research Letters, 47, e2020GL089576, https://doi.org/10.1029/2020GL089576, 2020a.

Wang, L., Kwiatek, G., Rybacki, E., Bonnelye, A., Bohnhoff, M., and Dresen, G.: Laboratory study on fluid-induced fault slip behavior: The role of fluid pressurization rate, Geophysical Research Letters, 47, e2019GL086627, https://doi.org/10.1029/2019GL086627, 2020b.

Wiemer, S. and Wyss, M.: Minimum magnitude of completeness in earthquake catalogs: Examples from Alaska, the Western United States & Japan, Bull. Seismol. Soc. Am, 90, 859–869, https://doi.org/10.1785/0119990114, 2000.

Zhou, H.: Rapid three-dimensional hypocentral determination using a master station method, Journal of Geophysical Research: Solid Earth, 99, 15439–15455, https://doi.org/10.1029/94JB00934, 1994.

---

## Author Comment (AC2) · 14 Dec 2020

**Response to Reviewer RC2 comments: se-2020-139-RC2**

We thank the anonymous reviewer RC2 for the thoughtful and positive review of our manuscript. The constructive comments helped us to further improve the manuscript.
We edited the manuscript carefully and addressed all comments of reviewer RC2. Please find below the detailed reply to the comments.

All reviewer comments are shown and highlighted as bold text, followed by our answers as indented normal text. Line numbers in our response refer to the tracked revised manuscript.

General comments of Reviewer RC2

**Seismicity associated with the stimulation of an Enhanced Geothermal System in Finland is presented. A variety of analytical tools are used to extract as much information as possible. While I do not have expert knowledge of the tools, they seem to have been competently used and deliver plausible results.**

**While I am not personally involved in EGS studies, I found the paper interesting because it relates to work that my group is doing with regard to seismicity induced by mining and the flooding of worked-out mines, as well as shale gas development.**

**I failed to meet the review deadline, for which I apologise. I downloaded a copy of the Manuscript supplementary material when I reviewed the paper on 14 November. I hope that the supplements are of the same high standard, but I have not checked this.**

> Authors:
> We apologize for the problem of downloading the supplements and hope that the Figures in the supplements also satisfy your expectations.

**Generally the paper is well-written. There are some minor grammatical errors that I have indicated on the attached annotated version of the manuscript. The referencing style is also inconsistent - some paper titles are in Sentence case, others in Title Case.**

> Authors:
> Thank you for indicating grammatical errors. We corrected all of them in the revised manuscript.

**There are also a few instances where I found the discussion difficult to follow or figures difficult to interpret. I have highlighted these and offered suggestions for improvement.**

> Authors:
> We address the comments and suggestions in detail below where each of them is listed and followed by our response.

Particular comments of Reviewer RC2

**1) Ln 225: '…Fig S2.' I was not able to view the supplementary figures.**

Authors:
We again apologize for the problem.

**2) Ln 233-235: "Two events with $M_W \geq 0.9$ occurred within the first 11 days of the post-stimulation phase. Two further $M_W > 1$ events occurred within 24 hours and 17 days after the stimulation ended, one with moment magnitude of 1.6 (Fig. 2)."**
**I am confused. Perhaps I do not really understand what you mean by 'after shut-in' and 'end of injection, 'bleed-off of wellhead pressure', 'post-stimulation phase'. After enlarging the graph, I count the seven events, three occurring just after the dashed line (bleed-off). I then see 3 events >= 0.9 in days 5-10.**

Authors:
Thank you for the hint that there are actually 3 events with $M_W \geq 0.9$ in days 5-10. We apologize for this mistake. Indeed, there are 3 events instead of only 2. We corrected the number in the manuscript.

**3) Ln 253-254: "…with two of them located on the NW flank of the injection well OTN-3…".**
**Figure 3a only shows one red rectanlge to NW of OTN-3. Is the second cluster the events that fall mostly in the cell defined by easting (-600;-400); northing (-200; 0)?**

Authors:
Yes, with the second cluster located at the NW flank of the injection well *OTN-3* we meant the clustered post-stimulation events that are located mostly in this cell.

**4) Ln 267-268: "The temporal evolution of the $CM_0$ separated for each hypocenter cluster is shown in Fig. 5."**
**Please make it absolutely clear to the reader where these three clusters lie. I suggest that you circle and label them in Figure 3.**

Authors:
We marked the three main hypocenter clusters by dashed rectangles in Fig. 3b and labeled them with the same names as we used in Fig. 5 to avoid any misunderstanding.

We further changed the following sentence in the manuscript (lines 281-282):

*"The temporal evolution of the $CM_0$ separated for each hypocenter cluster, marked in Fig. 3b, is shown in Fig. 5."*

**5) Ln 325: "…due to appearing ambiguities in…"**
**I am not sure what you mean by 'appearing ambiguities'. Why not just 'amiguities'?**

Authors:
The word "appearing" does not really explain ambiguities any further in this context. We therefore deleted the word "appearing" in the manuscript to not confuse the readers.

**6) Ln: 401: "…gravity of the cool water…"**
**It not clear what you mean here. Perhaps 'gravity-driven movement of the cool water into …'**

Authors:
Yes, the movement of the cool water into warm and less dense pore fluid would be driven by gravity. Thus, any further pressure would not be needed to migrate the water towards deeper parts of the reservoir. However, with obtaining new results (please see also responses to comments #15 and #16 of reviewer RC1), the statement is now more doubtful and thus, we deleted the sentence in the manuscript.

**7) Ln 434: "…lightened up…"**
**Not sure what you mean by 'lightened up'. Perhaps 'activated'.**

Authors:
Yes, with "lightened up" we mean "activated".

We changed the sentence in lines 452-453 as followed:

*"The 2018 seismicity activated a pre-existing network of small-scale parallel fractures dipping to ENE, in agreement with the dip direction of the inclined part of the injection well."*

**8) Ln 460: "…the gravitation-driven downwards migration…"**
**The physics behind the 'gravity-driven migration' is not clear to me. Is this related to the sinking of the cooler water?**

Authors:
The observation of the depthward migration of seismicity with time would be comparable e.g. with induced seismicity at The Geysers geothermal field (Kwiatek et al., 2015). For the Geysers, this depthward seismicity migration documents also a migration of cooler injected water into warmer pore fluid toward greater depth without any further pressure needed. The depthward migration of the water is also facilitated by steeply dipping faults which are well-known at The Geysers.

**9) Ln 463: "…but weak faults…"**
**Not sure what you mean by "but weak faults ..". Perhaps. These are thought to be weak faults ....**

Authors:
Yes, these fractures are thought to be weak faults.

We updated the sentence in the manuscript:

*"We conclude that seismic slip occurs on sub-parallel network of favorably oriented pre-existing but weak fractures, striking in NNW-SSE direction and dipping 45° ENE."*

**10) Ln 487-488: "…Seismic Moment Evolution During Hydraulic Stimulations,..".**
**Sentence case, not Title Case.**

Authors:
Thank you for the hint. We changed the reference into sentence case.

**11) Ln 518-519: "Hardebeck, J. and Shearer, P.: A New Method for Determining First-Motion Focal Mechanisms, Bulletin of the Seismological Society of America, 92, 2264–2276, https://doi.org/10.1785/0120010200, 2002."**
**I would expect the reference to follow Hardebeck and Michael.**

Authors:
Thank you for noticing this mistake. We swapped the reference of Hardebeck and Shearer (2002) with the reference of Hardebeck and Michael (2006).

Comments of reviewer RC2 to the Figures

**12) Figure 3. I find the display confusing. The colours used to show the seismic events correlate well with the legend; however, the colours on the trace of the well do not.**
**When I enlarge the Figure 3b I see a bright red tip; above it is an olive green section; and above that a bright green section, then a blue section, and finally a purple section. Are these five sections meant to correlate with P1 - P5?**
**I also see several red circles plotted on the well trace between -5500 and -5650 (and one at -5500). What do these signify?**

Authors:
We apologize for the confusion about the colored bands along the borehole trace in Fig. 3. Unfortunately, the colors along *OTN-3* were wrongly plotted. The colors should correlate with the colors of the five stimulation stages. Therefore, we updated the bands, now using the same colors as for P1-P5.

For a better visibility, we also changed the color of the stimulation phase P5 hypocenters to a darker yellow in Fig. 3.

We also apologize for the confusion about the small red circles. This was a mistake in potting. In the updated version of Fig. 3, we excluded these circles.

**13) Figure 4. Figure 2 indicates that seismicity was recorded for 65 days after the end of injection. As I read it, this figure only shows the cumulative moments for 30 days. Am I reading it correctly? If so, please note this in the caption and text.**

Authors:
Yes, this is correct, Fig. 4 only presents the time period of 30 days for each stimulation phase and indeed, the post-stimulation time period was 63 days long. However, we decided to not plot the full 65 days due to an insignificant increase in the cumulative seismic moment after 30 days of the end of injection.

We changed the following sentence in the caption of Fig. 4:

*"For a time period of 30 days, the temporal evolution of cumulative seismic moment release for the relocated seismicity is shown for each injection phase as well as for the post-stimulation phase."*

We also changed the following sentence in the text (lines 274-276):

*"Here, we show the temporal evolution of the cumulative seismic moment ($CM_0$) release for a time period of 30 days during the post-stimulation period and compare it with the evolution before shut-in of injection."*

**14) Figure 5. Please make it absolutely clear to the reader where these three clusters lie. I suggest that you circle and label them in Figure 3. Ensure that the time axis is marked so that it is clear to the reader that it covers the same duration as Figure 5 i.e. more than 10 days.**

Authors:
Thank you for the suggestion. To avoid any misunderstanding or confusion, we marked and labeled the three different clusters in the updated Fig. 3b, as already mentioned in the response of comment 4.

We also changed the x-axes in Fig. 5 to show and label the same time period as the x-axis in Fig. 4.

---

## Author Response (AR2)

Dear Michal Malinowski,

we thank you and the anonymous reviewer #1 for the second review of our manuscript. We are grateful for the constructive minor comments which helped us to even further improve the manuscript.

Please find below the detailed reply to the comments. All reviewer comments are shown and highlighted as bold text, followed by our answers as indented normal text. Line numbers in our response refer to the tracked manuscript.

If the manuscript gets accepted, we kindly ask you to allow us replacing the general data services link in the section *data availability* of the manuscript by the precise link that refers to our data publication and which is generated at the moment.

We appreciate your time and hope that our revised manuscript now qualifies for publication in Solid Earth.

Yours sincerely,

Maria Leonhardt

**Response to referee #1 comments**

Major comments of reviewer #1

**1) Ln 101,102: please indicate the minimum magnitude of the two data sets incl. 5456 and 55707 events. It would be also useful to mention what type of detection/picking was used for the two data sets.**

Authors:
We added the minimum moment magnitude of both event sets to the paragraph:

*"The reprocessed seismic catalog with description of its properties is available as separate data publication (see section data availability) and consists of 5,456 events with $M_W$ ≥ -2.47 that were detected and located during and after the stimulation (industrial monitoring) and reprocessed in our study. A total of 55,707 events with $M_W$ ≥ -0.95 were further detected during and after the stimulation but were not located or processed later on."*

Details of the type of detection/picking are mentioned in the description of our data publication. To avoid repetition, we decided to not mention them in the manuscript again.

**2) Ln 117: 'A total of 2,958 reprocessed events were located...': please specify what was the reason for relocation only about 60% of the catalog events: too large RMS, or position too far from the well or something else?**

Authors:
The hypocenter position of the events close to the injection well was the crucial factor for considering only ~60 % of the located 5,456 events for relocation and further analysis. We focused only on the 2,958 events that were located close to the injection

well *OTN-3* within a defined target volume at an epicentral distance of less than 5 km around the well and at depth of 4.5 to 7 km (as also mentioned in lines 118-120).

**3) Ln 212: Only the two shallower clusters elongate along SHmax, please be more precise.**

Authors:

To be more precise, we changed the sentence as follows:

*"Elongation of the two shallower clusters in SE-NW direction is sub-parallel to the local maximum horizontal stress $S_H^{max}$ = 110° (Kwiatek et al., 2019; Heidbach et al., 2016; Kakkuri and Chen, 1992)."*

**4) Ln 214: What do you mean by saying '…spans ~700 m depth. This exceeds 215 vertical relocation precision…'? I believe this is not surprising as your estimated relocation precision is 52m (Ln 129).**

Authors:

This is correct. We simply wanted to indicate that the thickness of the seismic cloud in vertical direction is a real feature.

**5) Ln 219 - 220: The statement 'The post-injection seismicity … seems to be mostly confined to three isolated clusters, with two of them located on the NW flank of the injection well OTN-3' is not very clear - I do not see two grey clusters at this position. Possibly you can omit this statament as it does not bring valuable info.**

Authors:

We agree, the statement does not bring enough information and therefore we deleted this statement and combined the first part of the sentence with the following sentence:

*"The post-injection seismicity shows no spatial migration and the largest post-stimulation events with magnitudes between $M_W$ 1.0 and $M_W$ 1.5 occurred at the NNW and SSE outer edge of the main cluster."*

We also deleted a similar sentence in the Discussion in lines 342-343.

**6) Ln 293: 'We further analyzed qualitatively the polarity pattern of events with polarities estimated from cross-correlation based technique of Shelly et al. (2016) ..': It is not clear whether this analysis is also shown in Fig. 8.**

Authors:

In Figure 8 we showed the most repetitive polarity pattern of all focal mechanisms, regardless if the focal mechanisms were obtained with manually picked polarities or with estimated polarities by the Shelly approach.

To avoid any confusion, we updated the sentence in lines 286-287 as follows:

*"Regardless of manually picked or estimated polarities, the most repetitive polarity pattern observed at each station for a particular family is plotted in Fig. 8a-c."*

**7) Ln 295 - 305: it is not easy to follow this section without illustrating the situation in a plot/table.**

Authors:
We included a table to the supplements (Tab. S1) which shows for each family and station how many of all events with focal mechanisms (in percent) display the same polarity pattern as the most repetitive polarity pattern presented by the focal mechanisms in Fig. 8a-c.

We further included the following sentence to the manuscript in lines 287-288:

*"For each family and station, the percentage of FM events showing this repetitive pattern is presented in Tab. S1."*

We further slightly updated the paragraph (lines 298-307) for a better understanding.

**8) Ln 377-382: The stress computations need be better explained. The point is that stress varies with depth and is thus unique for each event. And further, how do you estimate the pore pressure to get effective stress?**

Authors:
The stress at 6.1 km depth has been first derived in study of Backers et al. (2016), but we do not have depth-dependent stress profile. As indicated in the text, stress field orientation and stress shape ratio is very similar to that derived from application of BRTM method. Unfortunately, the amount of data disallows to discuss any spatial changes in the stress orientation. Thus, we focused on quantifying which fault plane families are more likely to fail first in the determined stress field (no presence of fluids). In the presence of the enhanced pore pressure, fault planes with higher instability coefficient (i.e. locate closer to failure envelope) would fail first. Such analysis requires, however, detailed data on spatial and temporal changes in the stress field, which is note possible, and also out of the topic actually discussed in this part of the manuscript. Text fragments in lines 314-322 and 382-389 have been modified to explain better the stress state and Mohr circle.

**9) Ln 384 - 390: Fig. S5 should be moved to the main part as Fig. 11 and Table S1 as well.**

Authors:
We included Fig. S5 and Tab. S1 to the manuscript as Fig. 11 and Tab. 1, respectively.

Minor comments of reviewer #1 on language

**10) Ln 279: 'very different in between families' - 'in' appears superfluous.**

Authors:
We deleted the word "in" in the sentence.

**11) Ln 280: between three -> among three.**

Authors:
Thank you for the hint. We updated the sentence by replacing "between" with "among".

**12) Ln 316: 'The stress inversion of the induced seismic events represents a local reverse faulting regime.' Stress inversion as a method (correctly Inversion of focal mechanisms for stress) cannot represent a faulting regime. I suggest 'The stress obtained by focal mechanism inversion represents...'**

Authors:
We changed the sentence as follows:

*"The stress obtained by focal mechanism inversion represents a local reverse faulting regime."*

**13) Ln 386: S5 should read as S6, same for Ln 393.**

Authors:
In the revised supplements, we did not include a Fig. S6 and therefore the references to Fig. S5 in Ln 386 and Ln 393 in the manuscript were correct. However, by now including Fig. S5 as Fig. 11 to the manuscript, we also updated the references in the manuscript.

Particular comments of reviewer #1 on Figures

**14) Fig. 2: To make the caption more informative, please give the number of events shown (2958?)**

Authors:
We included the numbers in the captions of Fig. 2:

*"The magnitudes of 2,958 absolute located and 1,986 relocated events are shown as grey and orange dots, respectively."*

**15) Fig. 3: Please indicate the profile of the depth section (b) in the map view (a).**

Authors:
We updated Fig. 3 showing now the profile of the depth section of Fig. 3b.

**16) Fig. 8: It would be useful to give the number of events for each family also in the caption, better than in the text**

Authors:
We updated the first sentence of the caption of Fig. 8 by including the number of events of each family:

*"Mean fault plane solutions (black lines) calculated from best FPSs of 99 events forming family I (a), 60 events forming family II (b) and 27 events forming family III (c)."*

**17) Fig. S3: The figure caption is not correct. According to your reponse 14) the light gray symbols show all events, not those with Mw>=1.**

Authors:
Thank you for the hint. We updated the first sentence of the caption as follows:

*"Map view of all relocated events."*

**18) Fig. 10: In the caption you state that 'A stress ratio of R = 0.53 was used for stress inversion.' This sounds confusing as R is the result of stress inversion.**

Authors:
To avoid any confusion, we changed the sentence in the caption of Fig. 10 as follows:

*"The stress inversion resulted in a stress ratio of R = 0.53."*

Particular comments of reviewer #1 on data publication

**19) In Fig. 5 you show the Gutenberg-Richter distribution of the new catalog. It would be however much more iknformative to show two curves: one for the original catalog and another for the new one. Then one would see at which magnitude levels most new events were detected.**

Authors:
The Gutenberg-Richter distribution of the original catalog is already presented in the supplementary materials of Kwiatek et al. (2019) and thus we refrain from showing this plot in our data publication again.